

# On *Dracograllus miguelitus* sp. nov. (Nematoda: Draconematidae) from an inactive structure: insights into its taxonomy, biodiversity and ecology at hydrothermal vents

William Johnson da Silva[1], Daniela Zeppilli[1], Valentin Foulon[2], Pierre-Antoine Dessandier[1], Marjolaine Matabos[1] and Jozee Sarrazin[1]

[1] Ifremer, BEEP, Univ Brest, Plouzané, France
[2] ENIB—École Nationale d'Ingénieurs de Brest, Plouzané, France

## ABSTRACT

**Background:** Hydrothermal vent fields are habitats to a diverse array of benthic organisms, including several nematode species, which represent a significant portion of the biodiversity in these environments. Despite their ecological importance, most research on hydrothermal vents has focused on macro-invertebrates. As a result, vent nematode biodiversity remains largely unexplored, especially in peripheral and inactive structures, underscoring the need for further investigation. A sampling program conducted in 2017 and 2018 along a gradient of venting activity led to the collection of a number of Draconematidae species in various habitats. In this article, we introduce *Dracograllus miguelitus* sp. nov., the first species of the genus described at a hydrothermal vent field, sampled from a visually inactive sulphide structure.
**Methods:** The samples were collected at the Lucky Strike vent field, on the Mid-Atlantic Ridge, using the suction sampler of the Remotely Operated Vehicle Victor6000. Specimens were retrieved from an edifice covered by a black layer of manganese oxy-hydroxides, with no local visible hydrothermal activity, at a depth of 1.639 m. Samples were sieved on a 32 μm mesh onboard, sorted and, for nematodes, identified to species level back in the lab. Fluorescent images were obtained using the ApoTome Fluorescence Microscope Module, and 3D observations were possible through the depth change method.
**Results:** We established *D. miguelitus* sp. nov. as a new species based on the combination of the following characters: four cephalic adhesive tubes (CATs), an elongated loop-shaped amphid with varying branch sizes between males and females, and a circular amphid in juveniles. Additionally, females display a minute setae emerging from the vulvar aperture. In males, the posterior adhesive tubes (PATs) are arranged in four longitudinal rows: two sublateral rows, each containing 10–12 PATs, and two subventral rows, consisting of 10 PATs in each. In females, sublateral and subventral rows with 13 PATs each. So far, *D. miguelitus* sp. nov. is the first species of the genus to be described from a hydrothermal environment and the deepest one. Beyond the formal description of this new species, we provide ecological and taxonomic backgrounds on Draconematidae at hydrothermal vents, with insights into the genus distribution, biogeography, and nomenclatural issues.

Corresponding author
William Johnson da Silva,
William.Johnson.Da.Silva@ifremer.fr

**Conclusion:** This discovery contributes to the knowledge of Draconematidae biodiversity, and highlights the importance to investigate nematode communities at species-level, data that is often missing at vent studies. Additionally, it underscores the significance of preserving inactive hydrothermal habitats, which are threatened by deep-sea mining activities.

## INTRODUCTION

A significant portion of the benthic diversity associated with hydrothermal vents is represented by nematodes, which play crucial roles at the ecosystem level such as bioturbation and organic matter degradation (*Vanreusel, Van den Bossche & Thiermann, 1997*; *Vanreusel et al., 2010a*, *2010b*). These environments are characterized by a hard substratum with high contents of metal compounds such as copper, zinc and iron, resulting from the precipitation of polymetallic sulphides contained in the vent fluids (*Hoagland et al., 2010*). Unlike other deep-sea ecosystems, hydrothermal vents exhibit a unique combination of low diversity and high biomass, largely driven by chemosynthetic energy sources (*Tunnicliffe, 1991*). Nematodes thrive in these conditions, highlighting their remarkable ability to adapt to habitat heterogeneity and extreme environments (*Vanreusel et al., 2010b*).

Surviving the peculiar environmental conditions of the deep sea, such as high pressure, low temperatures, and food scarcity, poses a significant challenge for faunal communities. Hydrothermal vents introduce further selective pressures linked to the type, origin, and intensity of hydrothermal activity and resulting environmental conditions (*Koschinsky et al., 2008*). Consequently, nematode communities at vents differ from those in the surrounding deep-sea. They harbor species that possess adaptations and strategies that are essential to survive in these harsh environments (*Vanreusel et al., 2010a*). Species diversity vary significantly across sites with different levels of hydrothermal activity (*Gollner, Miljutina & Bright, 2013*) and differences in species composition underscore their ability to occupy various niches, making them important contributors to the functioning of hydrothermal ecosystems (*Vanreusel et al., 2010b*).

Some examples of these adaptations can be observed in the Draconematidae family (*Filipjev, 1918*). These nematodes are easily recognizable by their S-shaped body morphology, which is common to most species. This unique shape has earned them colloquial names of "walking nematodes" or "dragon nematodes." In addition to their distinct morphology, many Draconematidae exhibit specialized structures that are closely tied to their locomotion and habitat use. Their cephalic (CATs) and posterior (PATs) adhesive tubes are linked to glands that secrete adhesive substances. These secretions allow them to "stick" parts of their bodies to the substratum, enabling alternative movements with intervals of "attachment and release" of both anterior and posterior body regions (*Stauffer, 1924*; *Cobb, 1929*; *Clasing, 1980*; *Tchesunov, 2014*). The Draconematidae family

comprises 16 genera and 89 valid species (*Nemys, 2024*), most of which are commonly found in coastal regions, typically associated with biological structures such as worm tubes, algae and coral reefs (*Decraemer, Gourbault & Backeljau, 1997*). The unexpected discovery of Draconematidae species in high abundances at hydrothermal vents was first reported in the Guaymas Basin on the East Pacific Rise (2,000 m water depth) by *Dinet, Grassle & Tunnicliffe (1988)*. Since then, additional records of the family in deep-sea habitats, including hydrothermal vents, have been reported. Several genera typical of deep-sea environments were collected, such as *Cephalochaetosoma* (syn. *Bathychaetosoma*) and *Dinetia* from the subfamily Draconematinae, as well as *Prochaetosoma* from the subfamily Prochaetosomatinae (*Kito, 1983*; *Decraemer, Gourbault & Backeljau, 1997*; *Rho, Kim & Min, 2007*; *Rho & Min, 2011*, and references therein). On the East Pacific Rise (EPR), *Dinetia* sp. were associated with *Bathymodiolus* mussel beds (*Flint et al., 2006*). Similarly, at the Lucky Strike vent field, along the northern Mid-Atlantic Ridge (MAR), both *Dinetia* and *Cephalochaetosoma* were associated with *Bathymodiolus* mussels (*Husson et al., 2017*). More recently, an experimental colonization study showed that *Cephalochaetosoma* represented between 76% and 90% of the nematode community on inorganic substrata deployed in intense vent emission areas (*Zeppilli et al., 2015*).

The genus *Dracograllus Allen & Noffsinger, 1978* represents the largest genus within the family, with 25 valid species (*Min et al., 2016*; *Nemys, 2024*), most of them reported in shallow waters, and, as for several Draconematidae species, associated with biogenic structures (*Verschelde & Vincx, 1993*). Even without apomorphic characters, the genus can be distinguished from other genera by several features, including a non-enlarged cuticle in the head region, the absence of bilateral cephalic acanthiform setae on the head capsule (except for *D. stekhoveni*), the absence of precloacal copulatory thorns, and the presence of paravulvar setae in some species (*Allen & Noffsinger, 1978*; *Decraemer, 1988*; *Decraemer, Gourbault & Backeljau, 1997*). Up to now, no species of *Dracograllus* had been formally described from deep-sea or hydrothermal habitats, as their distribution is generally limited to depths shallower than 100 m. However, several recent studies have reported *Dracograllus* specimens at greater depths (*Vanhove, Arntz & Vincx, 1999*; *Gad, 2009*; *Zeppilli et al., 2014*; *Spedicato et al., 2020*), although none have been formally described so far.

Most part of the vent ecological studies have focused on the microbial and macrofaunal compartments on active hydrothermal structures, neglecting the smaller meiofauna and also, the fauna from regions adjacent to the vents and inactive structures. However, although they received less attention, there is an increased interest in studying inactive vents, because they are the main target for mineral extraction (*Menini et al., 2023*). Recent studies have shown differences in faunal diversity between hydrothermally active and inactive habitats with a much higher diversity in the latter (*Cowart et al., 2020*). Few studies in the vent periphery have shown that nematode diversity extends outside the active zones (*Vanreusel et al., 2010b*). In this context, it becomes critical to better understand their diversity patterns in vent ecosystems including in their sphere of influence (*Levin et al., 2016*). Such knowledge is essential for developing environmental management plans to mitigate the impacts of deep-sea mining.

To assess meiofaunal benthic biodiversity associated with different vent environmental conditions, a sampling was carried out at 1,700 m depth at the Lucky Strike vent field in three contrasting habitats: an active vent site, a 'visually' inactive structure, and an area peripheral to venting activity (*Cowart et al., 2020*). In this study, we describe for the first time a new species of *Dracograllus* sampled from a deep-sea inactive sulfide structure, and supply updates on the taxonomy, ecology, and distribution of the genus. Additionally, we provide a dichotomous key to aid in the identification of *Dracograllus* species. Finally, we examine the implications of our results for the conservation of hydrothermal ecosystems, focusing on species composition, interactions and ecosystem functions in the context of the challenges posed by the mining industry.

## MATERIALS AND METHODS

### Study area and sampling collection

The Lucky Strike (LS) vent field is located in the northern part of the Mid-Atlantic Ridge (MAR), south of the Azores (Fig.1A), with a mean depth of 1,700 m (*De Busserolles et al., 2009*). LS consists of three volcanic cones that harbor over 25 active hydrothermal edifices surrounding a central lava lake (*Humphris et al., 2002*; *Ondreas et al., 2009*). Each active site-or edifice-is made of several smokers as well as patches of diffuse venting areas that extend in the periphery. To characterize the meiofaunal communities at the vent field scale, three habitat types were sampled: an active area, the periphery away from hydrothermal activity and a visually inactive edifice (Fig.1B).

The active habitat was located on the Montségur edifice (37°17.28′N, 32°16.53′W), in the southern region of LS, and consisted of cracks on a flat hydrothermal slab at the base of the edifice. The peripheral habitat was approximately 30 m from Montségur (37°17.28′N, 32°16.52′W), and covered by a few centimeters of sediments. Finally, the visually inactive structure, peripheral to the active Sintra edifice, lied about 400 m north of Montségur (37°17.48′N, 32°16.50′W), and consisted in an indurated sulfide structure covered at its base by a thin black layer of manganese oxy-hydroxides (Figs. 1C, 1D). Sampling was conducted during the 2018 Momarsat cruise (*Cannat, 2018*) using the suction sampler of the Remotely Operated Vehicle Victor6000. Neither the peripheral nor the inactive habitats exhibited visible hydrothermal activity or typical vent fauna. Once onboard, samples were sieved on 300 and 32 μm mesh sizes, and the fraction between 32–300 μm was preserved in 4% borax buffered formalin.

### Sample preparation and image acquisition

Nematodes were extracted from the sediment by the use of colloidal silica (Ludox), with specific gravity of 1.39 (*Pfannkuche & Thiel, 1988*). Specimens were fixed in formalin, and after the (*De Grisse, 1969*) protocol, they were transferred to glycerol and mounted onto permanent slides (*Somerfield & Warwick, 1996*). Drawings and measurements were made using a light microscope Leica DM 2500 LED with the aid of a drawing tube and a Leica DMC 4500 camera.

For the fluorescent observations, a Zeiss Axio Imager.Z2 microscope equipped with an Colibri.7 light, an ORCAFlash4.OLT (Hamamatsu, Hamamatsu-city, Japan) camera and a

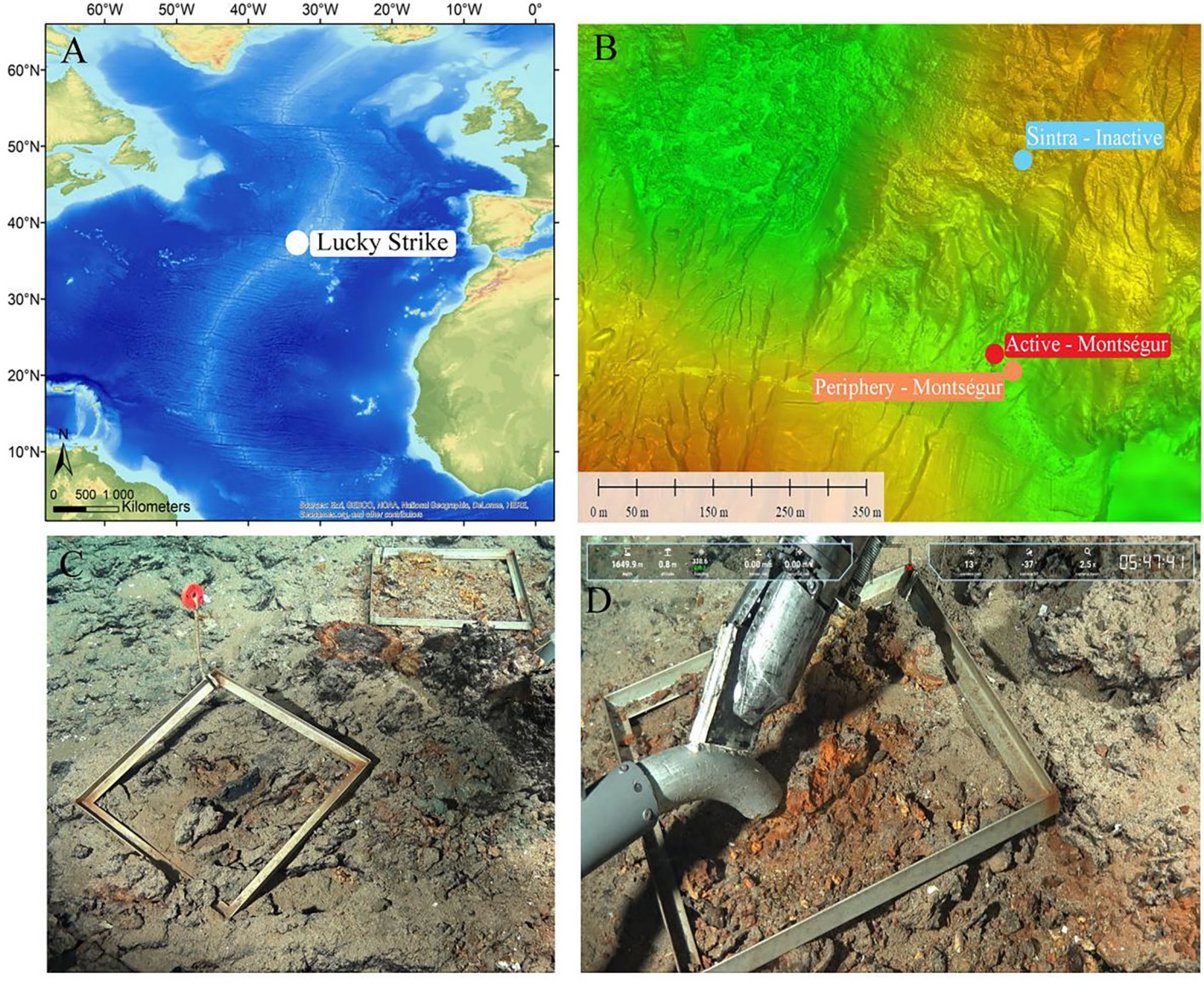

**Figure 1 Study site and sampling approach.** (A) Lucky Strike (LS) vent field along the Mid-Atlantic Ridge (MAR). (B) The three contrasting sampling sites. (C, D) Quadrats, faunal sampling and substratum view at the inactive habitat at LS. Source: Victor6000, Momarsat 2018, Ifremer. LS map modified from *Husson et al. (2017)*.

Apotome.2 slider module (for optical sections) was used. Autofluorescence and Phloxine B stain (Exitation 561 nm, Emission 571 nm) were used to observe internal and external structures in 3D. Thus, four fluorescent channels were used: Blue-filter Zeiss 49 DAPI Ex. G365 nm, Em. 445/50 nm, Green-filter Zeiss 38 HE GFP Ex 470/40 nm, Em LP 515 nm, Orange-filter Zeiss 43HE dsREd Ex. 550/25 nm Em. 605/70 nm, Red-filter Zeiss 50Cy5 Ex. 640/30 Em. 690/50 nm. Combinations of one to five channels (with brightfield) were used for optical section, increased depth of field and 3D depending on the observations. Images were processed using Zeiss Zen Pro and Arivis 4D Pro software.

In one of the earliest reviews of Draconematidae, *Allen & Noffsinger (1978)* provided key recommendations regarding specimen measurements, morphological analysis, and species delimitation. Building on their guidance, this study incorporates the following recommendations and observations: (1) measurements on the CATs should be taken on the right side of the nematode, (2), the length of the swollen esophageal and cephalic regions should be measured from the anterior tip of the lip region to just posterior to the swollen esophageal region (in most Draconematidae, the body is constricted in this region), (3) the body diameter width should be measured at the widest part of the swollen esophageal region while (4) the rostral width should be measured at the base of the rostrum, just anterior to the first body annule. For comprehensive details about measurements and possible variations along developmental stages, see *Allen & Noffsinger (1978)* and *Clasing (1980)*.

## Nomenclatural acts

The electronic version of this article in Portable Document Format (PDF) will represent a published work according to the International Commission on Zoological Nomenclature (ICZN), and hence the new names contained in the electronic version are effectively published under that Code from the electronic edition alone. This published work and the nomenclatural acts it contains have been registered in ZooBank, the online registration system for the ICZN. The ZooBank LSIDs (Life Science Identifiers) can be resolved and the associated information viewed through any standard web browser by appending the LSID to the prefix http://zoobank.org/. The LSID for this publication is: (AA6564D7-6BA7-405E-94D3-B659E62B8BDB). The online version of this work is archived and available from the following digital repositories: PeerJ, PubMed Central SCIE and CLOCKSS.

## RESULTS AND DISCUSSION

This is the first study to describe a new *Dracograllus* species from an inactive vent structure. It also corresponds to the greatest depth recorded among all known congener valid species. This finding provides insights into the diversity of potential habitats for Draconematidae, with genera and species distributed across a wide range of environments from shallow to deep regions.

### SYSTEMATICS
**Class CHROMADOREA** *Inglis, 1983*
**Subclass CHROMADORIA** *Pearse, 1942*
**Order Desmodorida** *De Coninck, 1965*
**Suborder Desmodorina** *De Coninck, 1965*
**Superfamily Desmodoroidea** *Filipjev, 1922*
**Family Draconematidae** *Filipjev, 1918*

**Diagnosis.** (Emended from *Leduc & Zhao (2016)*): Body short, S-shaped, usually with more or less enlarged pharyngeal and mid-body region. Cuticle annulated except for the head capsule (helmet) and tail terminus. Annules sometimes with spines, minute vacuoles, or a longitudinal lateral field in mid-body region or tail region. Cephalic sensilla in three

circles (6 + 6 + 4): six inner labial papillae, six outer labial setae, and four cephalic setae. Rostrum present (except in *Dinetia*). Amphideal fovea spiral to loop-shaped, rarely reduced or an internal longitudinal bar. CATs present, located dorsally on the cephalic capsule. Somatic setae arranged in eight longitudinal rows. Buccal cavity small to well developed, usually with a dorsal tooth, with or without subventral teeth. Pharynx cylindrical, dumbbell shaped, or with posterior bulb. Secretory–excretory system absent. At least the anterior-most PATs are arranged in four longitudinal rows, two subventrally and two ventrosublaterally, located on the posterior third of the body. PATs with or without differentiated tips, usually straight, rarely long and flexible. Copulatory apparatus with two spicules and trough-shaped gubernaculum. Three caudal glands extending beyond anus/cloaca.

## Subfamily Draconematinae Filipjev, 1918

**Diagnosis.** (Emended from *Decraemer, Gourbault & Backeljau (1997)*): Pharynx dumbbell-shaped. Swollen anterior body region is usually conspicuous and short. CATs with clearly open tips, mainly blister-shaped, and enlarged bases of insertion, located on the rostrum. Cephalic acanthiform setae may be present on the helmet. Buccal cavity narrow, unarmed. PATs shorts and straights (except for *Draconema trispinosum*, characterized by some longer tubes), with bell-shaped tips. All PATs are anterior to cloaca (except in *Dracograllus eira* and *Dracograllus minutus*). Copulatory thorns are rare. Paravulvar setae present or not. Tail cylindro-conoid with numerous nodules. Non-annulated tail region, with different length between species.

**Type genus.** *Draconema* Cobb, 1913.
**Genus *Dracograllus* Allen & Noffsinger, 1978**

**Diagnosis.** (Emended from *Leduc & Zhao (2016)*): Draconematidae, Draconematinae. Swollen anterior body region 18–26% of total body length. Usually eight CATs on the helmet, rarely 10 to 15. Non-enlarged cuticle on the rostrum. Amphids lateral, usually loop-shaped, with branches usually of unequal length, rarely spiral. Precloacal copulatory thorns are usually absent. PATs all anterior to cloacal opening (except in *Dracograllus eira Inglis, 1968* and *Dracograllus minutus Decraemer, 1988*), and usually extending posterior to anus in females (four exceptions). Paravulvar setae may be present.

**Type species.** *Dracograllus cobbi* Allen & Noffsinger, 1978.

**Remarks on *Dracograllus* taxonomy.** The genus *Dracograllus* is placed within the subfamily Draconematinae, along with *Draconema, Paradraconema*, and also the genus *Tenuidraconema*. On the other hand, the subfamily Prochaetosomatinae includes the deep-sea typical genera: *Bathychaetosoma, Cephalochaetosoma*, and *Dinetia*, as well as *Prochaetosoma*. This classification is based on characteristics such as the cylindrical shape of the pharyngeal sphincter with a terminal bulb, suggesting that the typical dumbbell-shaped pharynx was possibly secondarily lost in *Tenuidraconema* (*Decraemer, Gourbault & Backeljau, 1997*).

A molecular phylogenetic analysis of the family Draconematidae by *Rho & Min (2011)* revealed that the genus *Dracograllus* was the first lineage to diverge in the family, followed by five branching orders: *Dracograllus—Megadraconema—Draconema—Paradraconema* and *Prochaetosoma*. Consequently, the genus was excluded from the subfamily Draconematinae, as previously suggested. More recently, *Leduc & Zhao (2016)* examined the phylogenetic position of species within Desmodoroidea and confirmed the basal placement of *Dracograllus* within the Draconematidae, consistent with (*Rho & Min, 2011*) findings. Additionally, *Leduc & Zhao (2016)* found that sequences from *Dracograllus*, along with two other specimens of the genus, clustered with high posterior probability and bootstrap support, further reinforcing its basal position in the Draconematidae and providing new insights into the group's evolutionary relationships.

**Taxonomic issues.** Taxonomic issues within *Dracograllus* include synonymy, redescriptions, and taxon transfers, often arising from descriptions based on immature specimens or discrepancies in the number of longitudinal PAT rows. For example, *Dracograllus eira* was originally described as *Draconema eira* by *Inglis (1968)*, later synonymized with *Dracograllus eira*. Similarly, *Chaetosoma falcatum Irwin-Smith, 1918* underwent multiple reclassifications before being recognized as *Dracograllus falcatus Allen & Noffsinger, 1978*. Another synonym is *Tristicochaeta falcata* Johnston, 1938. Since *Allen & Noffsinger (1978)*, and now, the valid name is *Dracograllus falcatus*. See valid species and nomen nudum section.

*Allen & Noffsinger (1978)* first described *Dracograllus filipjevi* from holdfasts of kelps from Japan (Oarai, Ibaraki-ken, Honshu Island). It was characterized by: (1) larger body size (500–700 μm long), (2) the absence of cephalic acantiform setae, (3) the presence of longitudinally areolated body cuticle with dot-like punctations, (4) the presence of some cuticular collar in swollen pharyngeal region, and 9 sublateral and 8–9 subventral PAT in males, and 12–13 and 9–10 in females. *Rho, Kim & Kim (2006)* also found *D. filipjev* associated with calcareous algae in Daebo-ri, Guryongpo, Korea, at 3–5 m depth. However, the Korean specimens did not align well with the original description in the number of PATs in male with eight to nine tubes, compared to nine to 11 in the original description. Given that these characteristics are crucial for the taxonomy of Draconematidae, this discrepancy supported the redescription made by *Rho, Kim & Kim (2006)*.

Analyzing the Draconematidae from Guryongpo (Daebo-ri, Korea), *Rho & Min (2011)* reported several species of the genus *Dracograllus*. However, these species are considered invalid, meaning they do not comply with certain taxonomic criteria required for formal recognition (see Article 16.1 of the International Code of Zoological Nomenclature). According to this article, every new name published after 1999 must clearly indicate its new status using specific terms such as 'fam. nov.,' 'gen. nov.,' 'sp. nov.,' 'ssp. nov.,' or an equivalent expression (*e.g.*, 'species nova,' 'new species').

**List of valid species.**
*Dracograllus antillensis Decraemer & Gourbault, 1986*
*Dracograllus chitwoodi Allen & Noffsinger, 1978*

*Dracograllus cobbi* *Allen & Noffsinger, 1978*

*Dracograllus cornutus* *Decraemer, 1988*

*Dracograllus demani* *Allen & Noffsinger, 1978* [*Decraemer, 1988*; *Verschelde & Vincx, 1993*]

*Dracograllus eira* (*Inglis, 1968*) *Allen & Noffsinger, 1978*; [*Decraemer, 1988*; *Verschelde & Vincx, 1993*]

   Syn. *Draconema eira* *Inglis, 1968*

*Dracograllus falcatum* (*Irwin-Smith, 1918*)

   Syn. *Chaetosoma falcatum* *Irwin-Smith, 1918*

   Syn. *Notochaetosoma falcatum* (*Irwin-Smith, 1918*) *Cobb, 1929*

   Syn. *Drepanonema falcatum* (*Irwin-Smith, 1918*) *Cobb, 1933*

   Syn. *Claparediella falcatum* (*Irwin-Smith, 1918*) *Filipjev, 1934*

   Syn. *Draconema falcatum* (*Irwin-Smith, 1918*) *Kreis, 1938*

   Syn. *Tristicochaeta falcata* (*Irwin-Smith, 1918*) *Johnston, 1938*

   Syn. *Dracograllus filipjevi* *Allen & Noffsinger, 1978*

   Syn. *Dracograllus gerlachi* *Allen & Noffsinger, 1978*

   Syn. *Dracograllus gilbertae* *Verschelde & Vincx, 1993*

*Dracograllus grootaerti* *Decraemer, 1988*

*Dracograllus kreisi* *Allen & Noffsinger, 1978*

*Dracograllus laingensis* *Decraemer, 1988*

*Dracograllus mawsoni* *Allen & Noffsinger, 1978*

*Dracograllus minutus* *Decraemer, 1988* [*Gourbault & Decraemer, 1992*]

*Dracograllus miguelitus* **sp. nov.** *Johnson da Silva et al., 2025*

*Dracograllus ngakei* *Leduc & Zhao, 2016*

*Dracograllus papuensis* *Decraemer, 1988*

*Dracograllus pusillus* *Decraemer, 1988*

*Dracograllus solidus* (*Gerlach, 1952*) *Allen & Noffsinger, 1978*

   Syn. *Draconema solidum* *Gerlach, 1952*

*Dracograllus spinosus* *Decraemer, 1988*

*Dracograllus stekhoveni* *Allen & Noffsinger, 1978*

*Dracograllus timmi* *Allen & Noffsinger, 1978* [*Gourbault & Decraemer, 1992*]

*Dracograllus trispinosum* (*Allen & Noffsinger, 1978*) *Decraemer, 1988*

Syn. *Dracotoranema trispinosum* *Allen & Noffsinger, 1978*

*Dracograllus trukensis* *Min et al., 2016*

*Dracograllus wieseri* *Allen & Noffsinger, 1978*

**Nomen nudum**.

*Dracograllus brevitubulus Rho & Kim, 2011* (unaccepted > nomen nudum)

*Dracograllus geomunensis Rho & Kim, 2011* (unaccepted > nomen nudum)

*Dracograllus gosanensis Rho & Kim, 2011* (unaccepted > nomen nudum)

*Dracograllus jaewani Rho & Kim, 2011* (unaccepted > nomen nudum)

*Dracograllus jongmooni Rho & Kim, 2011* (unaccepted > nomen nudum)

*Dracograllus sungjooni Rho & Kim, 2011* (unaccepted > nomen nudum)
*Dracograllus chiloensis Clasing, 1980* (uncertain > taxon inquirendum)

**Description of *Dracograllus miguelitus* sp. nov.**
(Table 1; Figs. 2–5; S1–S4)

**Type material.** All specimens are deposited in the Muséum National d'Histoire Naturelle de Paris, France. Male holotype, two juvenile paratypes and the female paratype in the inventory number MNNH-BN511-I1-L1-B. Two male paratypes and female paratypes in MNHN-BN511-I2-L1-A.

**Other material.** Other specimens are held in the collection of the Laboratoire Environnement Profond of the Biologie et Ecologie des Ecosystèmes marins Profonds research unit-Ifremer, Plouzané, France.

**Etymology.** The specific epithet is in honor of 'Pedro Miguel', nephew of the first author.

**Type locality and habitat.** Lucky Strike vent field-MAR. Samples were collected from a hard substratum covered by a thin layer of volcaniclastic sediment, on a visually inactive vent structure at 1,649 m depth. Environmental conditions exhibited background or slightly higher seawater temperature (*i.e.* 4.8–5.7 °C) and higher pH (*i.e.* 7.8–7.9) than the surrounding deep-sea but the activity was very low compared to active habitats where recorded temperature varied between 5.2 °C to 9.5 °C, reaching a maximum of 22.1 °C, and pH varying from 7.2 to 7.6.

**Measurements.** Table 1.

**Holotype male.** Habitus typical for the genus. A total of 612 µm long, swollen anterior body region representing 19% of total length (Figs. 2A, 2C). Amphid elongate loop-shaped with non-equal branch sizes, and with one more ventrally than another, amphideal fovea 7.1 µm (Figs. 2C and 3C). Helmet strongly cuticularized (Fig. 2D), with punctations and granular appearance in the lateral part (Fig. 3A). Annulation without ornamentation along the body, except for the tail tip and helmet, with minute punctations (Figs. 2C, 2H, 3D). Four CATs on the rostrum (22.5–26.0 µm long), arranged dorsally in two transverse rows, all with enlarged bases (Figs 2C, 3A, 3B). Some specimens exhibited depressions resembling CAT insertions; however, none of these depressions contained tubes. The setae in the cephalic region and along the body possess a cuticular collar at their insertion and alternation of short and long setae, this collar is projected outside of the cuticle, as a pedicel setae (PS), with 1.2–2.1 µm long (Figs. 2C, 3A, 3B), also in the paratype male (Fig. 6E). Cephalic and cervical region with eight longitudinal rows of setae on each side, between 26 and 32 µm long, and also some irregular minute setae (6–12 µm long) (Fig. 2C). Slender cervical region without lateral differentiation (Figs. 2C and 3A). Buccal cavity narrow, unarmed (Fig. 2B). Cardia short. Pharynx dumbbell-shaped with a weakly developed isthmus from the muscular posterior large endbulb (Figs. 2D, 3A, 3B). Intestine narrow, mostly cylindrical, with a granular appearance, gradually widening posteriorly and lying dorsally to the reproductive system (Figs. 3B–3D). Reproductive system with a single and

**Table 1 Morphometric measurements (µm) of *Dracograllus miguelitus* sp. nov.**

| Parameter | Males | | Females | | Juveniles | |
|---|---|---|---|---|---|---|
| | Holotype | Paratypes (*n* = 2) | Paratype | Paratypes (*n* =2 ) | Paratype J3 | Paratype J4 |
| L | 612 | 630–735 | 765.5 | 748–788 | 426.3 | 514.3 |
| *a* | 10.6 | 13.7–14.9 | 11.6 | 12.0–12.8 | 14.2 | 10.1 |
| *b* | 6.9 | 7.4–7.5 | 7.3 | 7.7–8.1 | 5.7 | 6.3 |
| *c* | 6.2 | 6.2–7.0 | 8.1 | 8.4 | 6.4 | 6.9 |
| *c'* | 4.4 | 4.7–5 | 4.5 | 3.8–4.1 | 3.9 | 4.4 |
| Head diam.* | 32.4 | 27.2–33.92 | 31.1 | 29.6–34.37 | 19.3 | 24.3 |
| Amphid. length | 15.5 | 14.2–15.2 | 14.4 | 12.7–13.5 | 9.2 | 10 |
| Amphid. width | 7.12 | 6.8–6.9 | 8.2 | 7.5–7.9 | 4 | 4.1 |
| Amphid./cbd(%) | 22.0 | 20.4–25.0 | 26.6 | 25.4–25.5 | 16.7 | 20.1 |
| Amphid from ant. | 5.04 | 4.1–5 | 3.5 | 3.3–3.7 | 1.5 | 1.9 |
| Phar. length | 88.5 | 84.0–99.5 | 93.3 | 92–103 | 75.3 | 80.6 |
| Phar. bulb diam. (ant.) | 21.6 | 20.3–22.8 | 24.2 | 26.1–26.7 | 20.8 | 22.5 |
| Phar. bulb diam. (post.) | 30 | 29.3–33.5 | 36.7 | 34.1–38.5 | 24.2 | 28 |
| Max. body diam. Phar. | 57.4 | 45.7–56.3 | 58.8 | 58.4–61.6 | 40.6 | 54.3 |
| Max. body diam. Mb | 44.9 | 44.8–49.3 | 49.9 | 61.8–70 | 30.1 | 41.4 |
| Min. body diam. | 11.7 | 16.5–18.59 | 20.8 | 20.1–25 | 20.2 | 20.6 |
| Spic. length | 50.2 | 47.1–54.9 | – | – | – | – |
| Gub. Apoph. length | 13.6 | 12.7–14.5 | – | – | – | – |
| abd | 22 | 19.9–22.2 | 18.3 | 19.7–22.9 | 17.1 | 17.3 |
| T. length | 97.2 | 100.8–105.2 | 84.2 | 88.6–94.3 | 66.6 | 75.9 |
| Non. ann. T. length | 43.0 | 41.1–44 | 47.4 | 46.9–53 | 36.7 | 38.5 |
| Non. ann. T. length % | 44.2 | 40.8–41.8 | 56.3 | 56–59 | 50.7 | 55.2 |
| T. length/abd | 4.4 | 4.7–5.0 | 4.5 | 3.8 | 3.8 | 4.3 |
| Longest tail setae | 48.2 | 49 | 47.0 | 45.1–45.8 | 42.8 | 44.6 |
| CATn | 4 | 4 | 4 | 4 | 3 | 4 |
| CATl | 22.5–26.0 | 23.1–30.7 | 23.7 | 23.88–24.73 | 15.5 | 21.2 |
| 1SlATl | 62.5 | 63.4–68.4 | 58.9 | 56.8–58.8 | 47.4 | 50.4 |
| SlATn** | 10 | 10–12 | 13 | 13 | 5 | 7 |
| 1SvATl | 50.0 | 53.3–54.3 | 51.5 | 50.9–50.2 | 42 | 44.3 |
| SvATn** | 10 | 10 | 13 | 13 | 5 | 7 |
| V. to ant. | – | – | 317.1 | 298.5–302.4 | – | – |
| V. (%) | – | – | 41.4 | 39.9–41.3 | – | – |
| PS length | – | – | 5.0–5.9 | 5.5–6.1 | | – |
| V. b. diam. | – | – | 65.3 | 64.7–68.8 | – | – |

**Notes:**
J3 and J4, third-and fourth-stage juveniles, respectively. L, body length; *a*, ratio of body length to maximum body width; *b*, ratio of body length to pharynx length; *c*, ratio of body length to tail length; *c'*, ratio of tail length to anal body diameter; Head diam., head diameter*; Amphid. length, amphideal length; Amphid. width, amphideal width; Amphid./cbd (%), percentage of amphideal length relative to the corresponding body diameter; Amphid from ant., distance from amphid to anterior end; Phar. length, pharynx length; Phar. bulb diam. (ant.), anterior pharyngeal bulb diameter; Phar. bulb diam. (post.), posterior pharyngeal bulb diameter; Max. body diam. Phar., maximum body diameter at pharynx level; Max. body diam. Mb, maximum body diameter at mid-body; Min. body diam., minimum body diameter; Spic. length, spicule length (measured along the median line); Gub. Apoph. length, gubernaculum apophysis length; abd, anal body diameter; T. length, tail length; Non. ann. T. length, non-annulated tail tip length; Non. ann. T. length (%), percentage of non-annulated tail tip relative to total tail length; T. length/abd, ratio of tail length to anal body diameter; CATn, number of cephalic adhesive tubes; CATl, length of cephalic adhesive tubes; 1SlATl, length of first sublateral adhesive tube; SlATn, number of sublateral adhesive tubes**; 1SvATl, length of first subventral adhesive tube; SvATn, number of subventral adhesive tubes**; V. to ant., distance from vulva to anterior end; V. (%), vulva position as percentage of total body length; PS length, paravulvar setae length; V. b. diam., vulvar body diameter.
* Measured at amphid level.
** In each row.

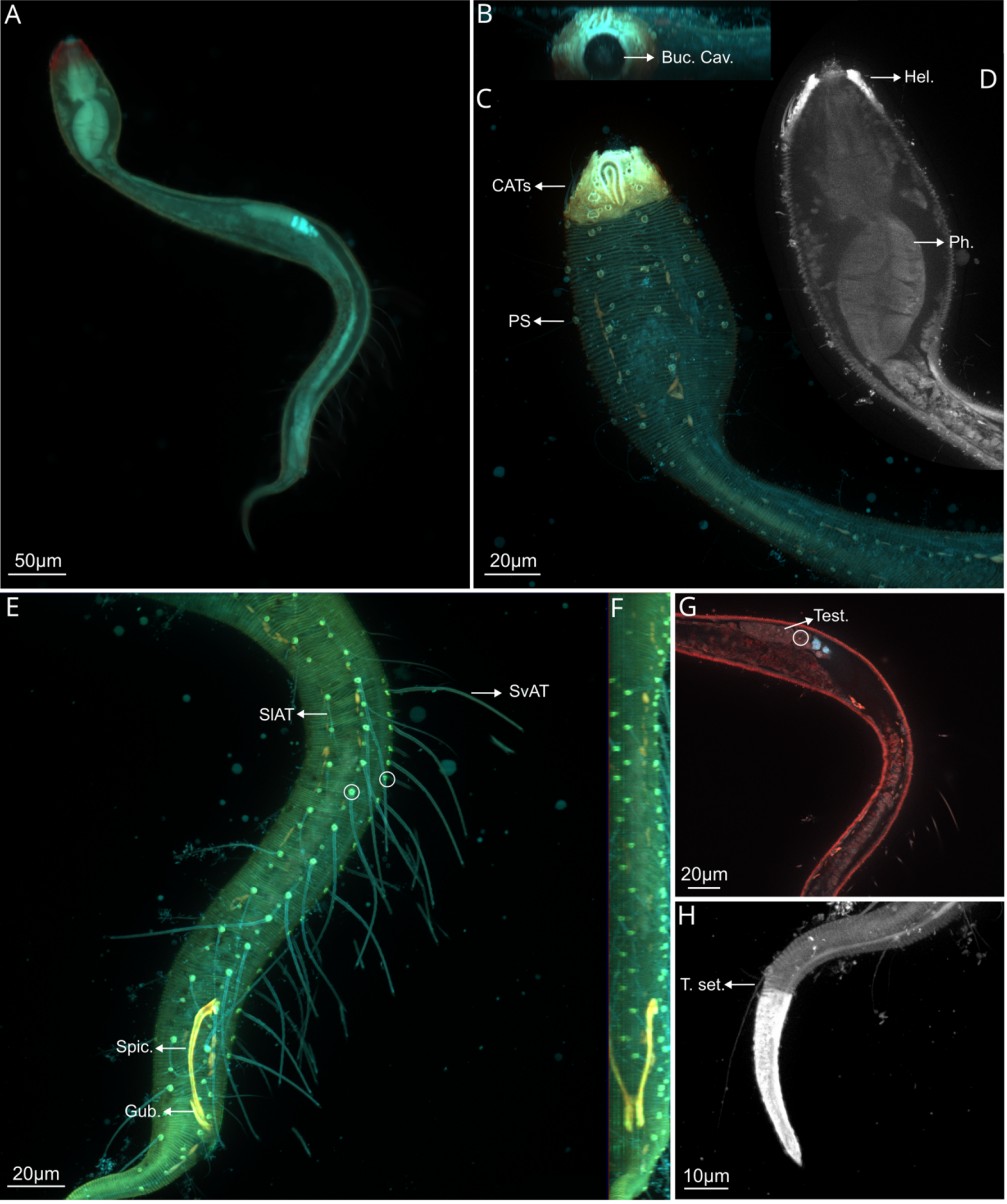

**Figure 2 *Dracograllus miguelitus* sp. nov., 3D fluorescence microscopy. Male holotype.** (A) General view (blue, green and red fluorescent channels). (B) Anterior view of the buccal cavity (blue and red fluorescent channels—maximum intensity projection). (C) Swollen anterior region (blue, green and red fluorescent channels—maximum intensity projection), showing the CATs, the amphideal fovea and four of the longitudinal rows of setae and the pedicel setae. (D) Internal view of the head region (blue fluorescent channel—optical section), with the well-cuticularized helmet, some of the CATs and the pharynx. (E) Posterior male region (blue, green and orange fluorescent channels—maximum intensity projection), with both sublateral and subventral rows of PATs, their insertion (circles), spicule and gubernaculum. (F) Ventral view of the posterior regions (blue, green and orange fluorescent channels—maximum intensity projection), with the arcuate spicules. (G) Mid-mody (blue and red fluorescent channels—optical section) showing intestine and testis, circle indicates reproductive cells. (H) Posterior tail region (blue fluorescent channel—maximum intensity projection), with the non-annulated tail region, and the setae associated. Arrows/Abrev: Buc. Cav, buccal cavity; CATS, cephalic adhesive tubes; PS, pedicel setae; Hel, helmet/cephalic capsule; Ph, pharynx; SlAT, sublateral adhesive tubes; SvAT, subventral adhesive tubes; Spic, spicule; Gub, gubernaculum; Test, testis; T. set, tail setae.

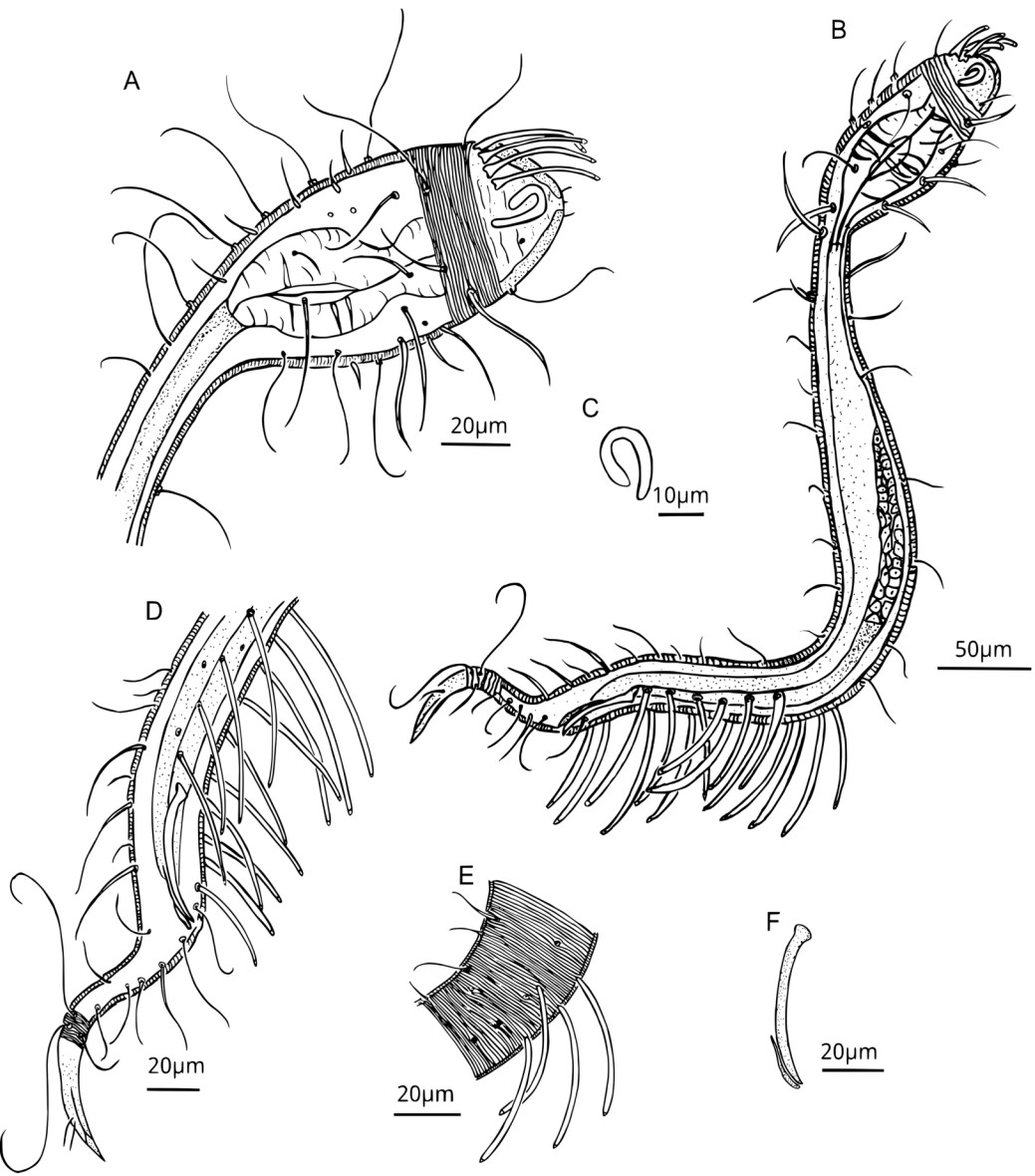

**Figure 3 *Dracograllus miguelitus* sp. nov. Male holotype.** (A) Head. (B) General view. (C) Fovea. (D) Posterior region, rows of PATs, not all tubes included. (E) Mid-mody curvature, showing the first PATs. (F) Cervical cuticle regions and the three first SvATs and first SlAT, spicule and gubernaculum. Abbrev: PATs, posterior adhesive tubes, SlAT, sublateral adhesive tubes; SvAT, subventral adhesive tubes.                               

outstretched anterior testis (monorchic) with a well-developed germinative region (Figs. 2G and 3B). Spicules 50.2 μm long, moderately arcuate (in some specimens more arcuate than in others), proximal region with an offset knob-like capitulum (Figs. 2E, 2F and 3F). Gubernaculum 13.6 μm long, lying parallel with the distal end of the spicules, with a minute distal and lateral wing-like expansions (Fig. 3F).

PATs weakly slender, with tongue-like tips (Figs. 2E, 3B, 3D and 3E), difficult to observe due to their thickness. All PATs located anterior to the cloacal opening (Figs. 2E, 3D).

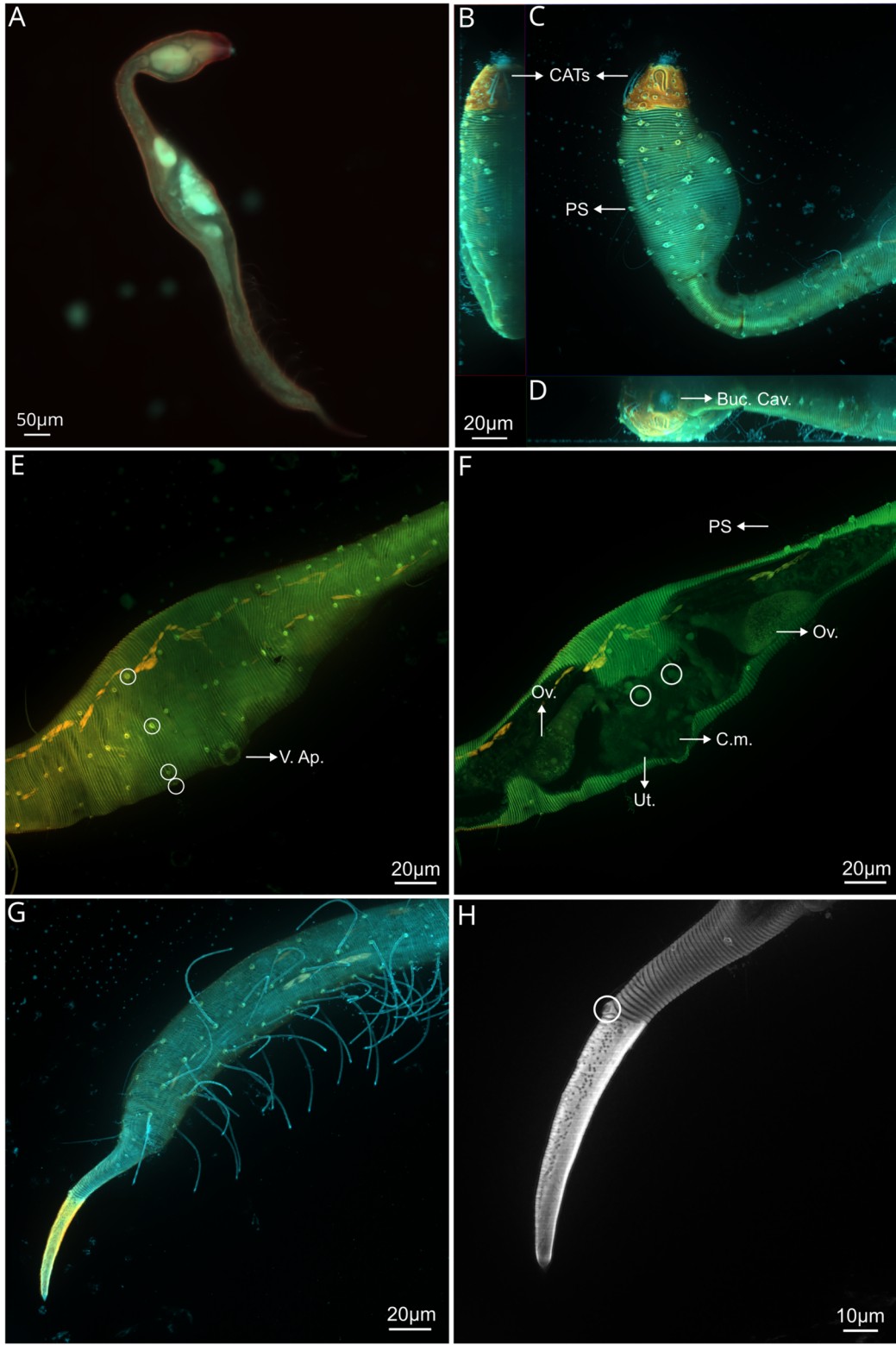

**Figure 4 Dracograllus miguelitus sp. nov., 3D fluorescence microscopy. Female paratype.**
(A) General view (blue, green and red fluorescent channels. (B) Dorsal view of the head (blue, green and orange fluorescent channels—maximum intensity projection), with the four CATs. (C) Head and cervical region (blue, green and orange fluorescent channels - maximum intensity projection), with evident

**Figure 4 (continued)**
fovea and pedicel setae, note the clear helmet ornamentation. (D) Face view of the buccal cavity (blue, green and orange fluorescent channels—maximum intensity projection), also CATs and fovea. (E) Mid body region (green and orange fluorescent channels—maximum intensity projection), with the vulvar aperture, and four of the longitudinal rows of setae (circles). (F) Internal view of female reproductive system (green and orange fluorescent channels—maximum intensity projection), reproductive cells within circles. (G) Posterior body region (blue, green and orange fluorescent channels—maximum intensity projection), with some of the both sublateral and subventral rows. (H) Posterior tail region (blue fluorescent channel—maximum intensity projection), with the non-annulated tail region and a setae insertion (circle). Arrows/Abbrev: CATs, cephalic adhesive tubes; PS, pedicel setae; Buc.Cav., buccal cavity; V. Ap., vulvar aperture; Ov., ovaries; Ut., uterus; C.m., constrictor muscles.

PATs are arranged in four longitudinal rows: two sublateral rows each with 10–12 PATs, and two subventral rows consisting of 10 PATs each one (Figs. 2E, 3D), intermingled setae are present between the PATs. Dorsal posterior part with small and irregular distributed setae between the setae following the rows of the body, more visible than in the ventral side, where only few and minute intermingled setae are present (Figs. 3D). Two pairs of setae are present in the annulated tail region. One pair of long setae in the last annules of the dorsal part of the tail, each setae with a collar at the base, close to each other (48–50 μm long), ventrally, one pair of shorter setae, with collar at the basis (11–12 μm long; Figs 2H, 4B-–4D). Additional five setae are present in the tail, in each somatic row of setae. One to two minute setae in the middle of the non-annulated tail tip (1.5 μm long), in some specimens, some of the setae were absent, but the minute insertion of them is visible.

**Paratype female.** Similar to males in most aspects, but differ in the following characteristics: greater length, with 765.5 μm long (748–788 μm long in paratypes female and 630–735 μm long in paratypes males; Fig 4A). Amphid also elongated loop-shaped, however slightly wider than in males. In addition, amphid in females is positioned more anteriorly than in males (Figs. 4C, 5B). Pedicel setae at both lateral and ventral side of the head, distributed for all body length and more developed than in males (2.5–3.2 μm long; Figs. 4C, 5B). Swollen pharyngeal region 18% of total body length. Some of the setae appear to be lost, with only the cuticular collars remaining, distributed irregularly and in smaller various sizes than in the head (Figs. 6A, 6B). Just some collars, without setae, are also present in the swollen head region and also on the helmet. Both the anterior and posterior regions of the pharynx vary between sexes. The anterior pharyngeal bulb in females has a diameter of 24–26 μm, and the posterior bulb has a diameter of 34–38 μm (compared to 20–22 μm long and 29–33 μm long respectively in males).

Reproductive system didelphic-amphidelphic with reflexed ovaries, both located ventrally relative to the intestine. Uterus filled with a mass of ovoid reproductive cells (Fig. 4F, circles). The region surrounding the vulvar aperture protrudes outward, with the cuticle giving a labial appearance (Fig. 4E). Two pairs of paravulvar setae present, one anterior and one posterior to the vulvar aperture, with length between 6.09–6.69 μm long. Also a setae emerging from the vulvar aperture (5.03 μm long; Figs. 5A and 5E). Well-developed contractor muscles in the vagina (Fig. 4F). PATs all anterior to the anus,

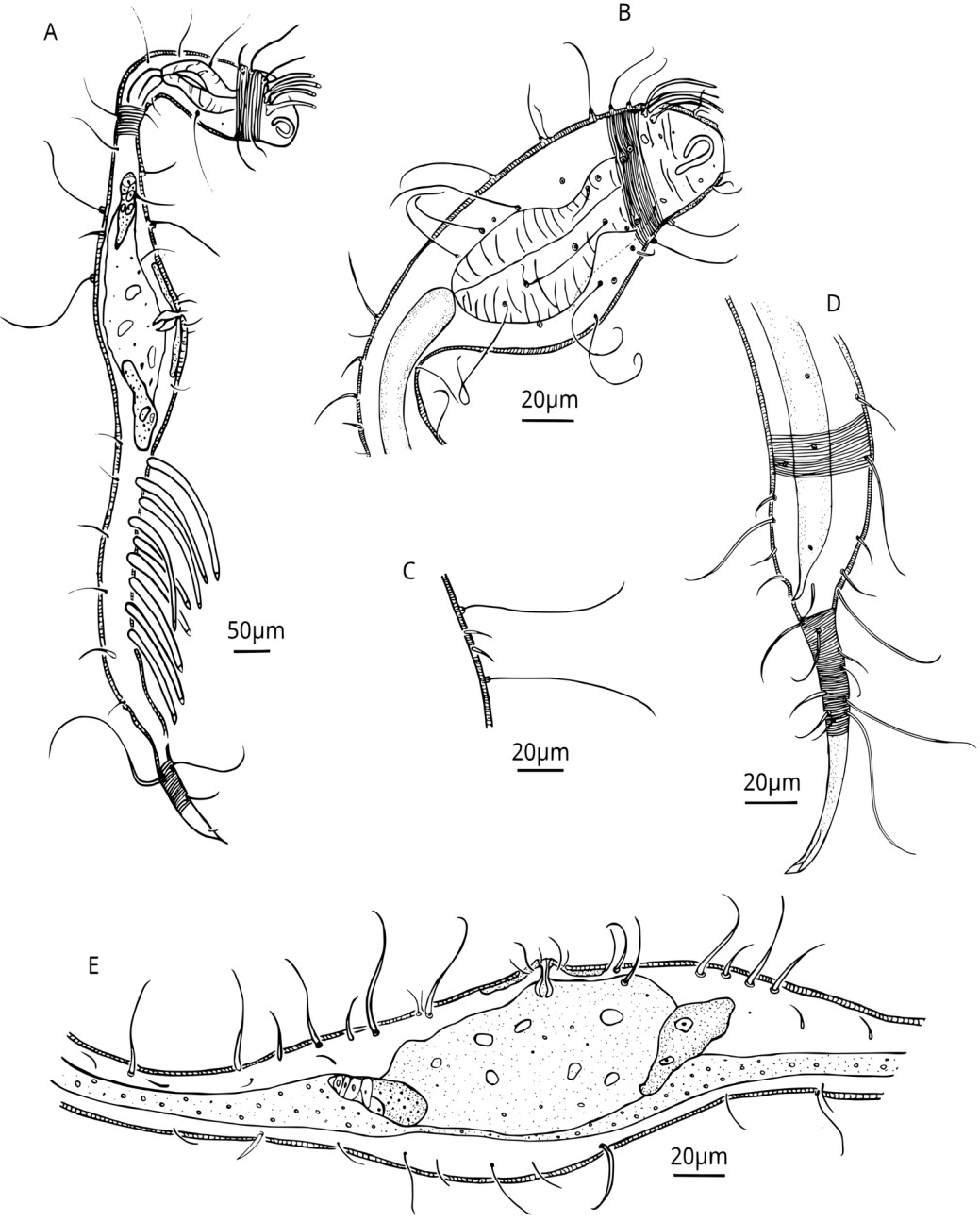

**Figure 5** *Dracograllus miguelitus* **sp. nov. Female paratype.** (A) General view; (B) Head. (C) Cuticle at cervical region. (E) Posterior and tail region, female reproductive system.

more slender, but shorter than in males (56.8–58.8 *vs* 63.4–68.4 in males), with weakly developed bell shaped tips with a tongue-like valve. PATs arranged in four longitudinal rows: two sublateral rows each consisting of 13 adhesion tubes with intermingling and irregular somatic setae and two subventral rows of 13 adhesion adhesion tubes also with intermingling somatic setae (Fig. 4G). First SlAT on the females with 58.9 μm long and

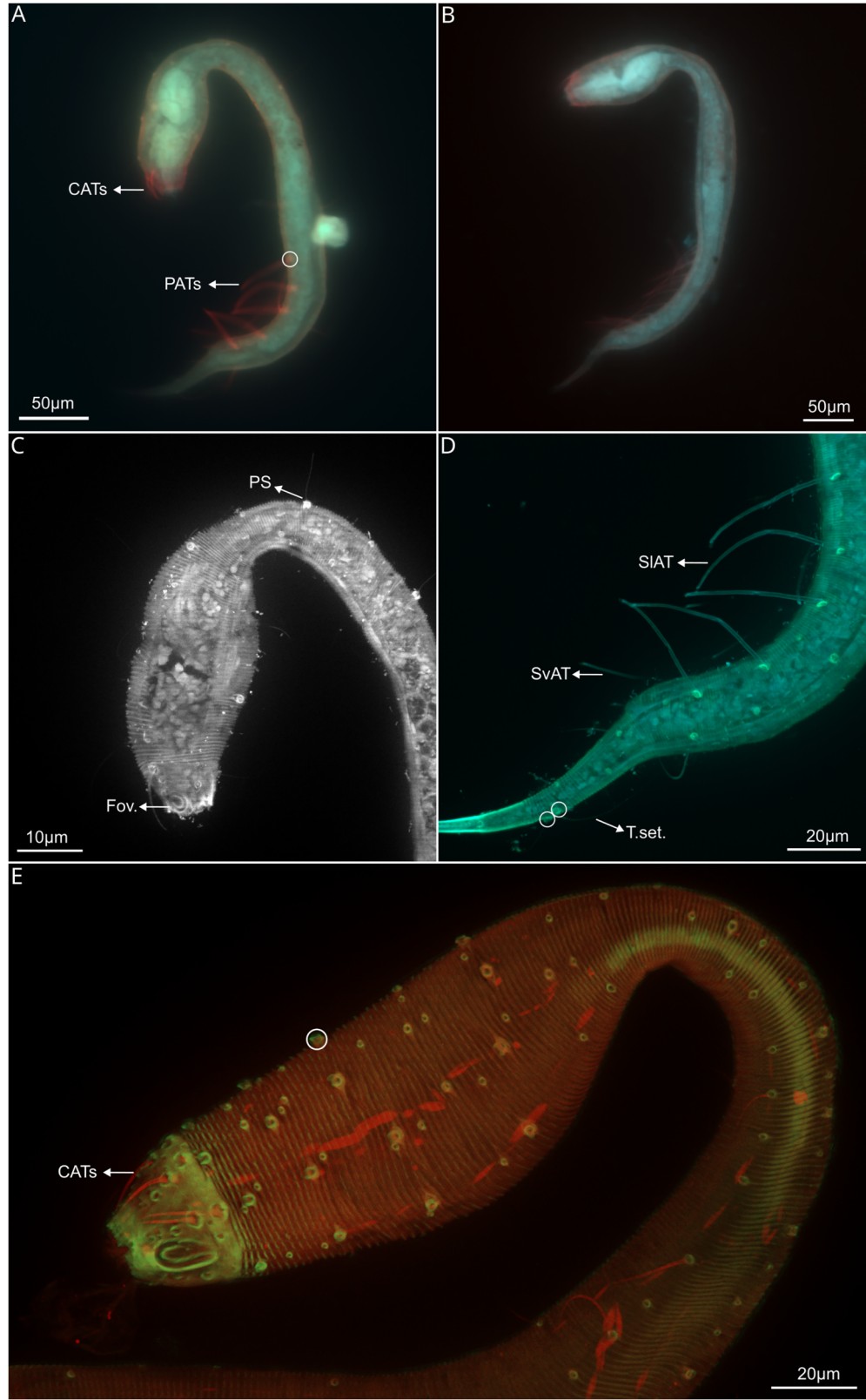

**Figure 6 *Dracograllus miguelitus* sp. nov., 3D fluorescence microscopy of the juveniles and a male paratype.** (A) General view of juvenile third stage (blue, green and red fluorescent channels), showing

**Figure 6** (continued)
both cephalic and posterior adhesive tubes. (B) General view of the juvenile fourth stage (blue, green and red fluorescent channels). (C) Head and cervical region of juvenile third stage (blue fluorescent channel—maximum intensity projection), arrows indicate the closed shape of fovea and the pedicel setae. (D) Posterior and tail region of the third-stage juvenile (blue and green fluorescent channels—maximum intensity projection, with arrows indicating the rows of adhesive tubes and circles highlighting the insertion points of the tail setae. (E) Head and cervical region of the paratype male (green and red fluorescent channels—maximum intensity projection), CATs on arrows and pedicel setae within the circle. Arrows/Abbrev: CATs, cephalic adhesive tubes; PATs, posterior adhesive tubes; PS, pedicel setae; Fov., fovea; SlAT, sublateral adhesive tubes; SvAT, subventral adhesive tubes; T.set., tail setae.

62.5 on males. All PATs weakly broadened at insertion base. Tail gradually tapering posteriorly to a cylindro-conoidal non-annulated tail tip (Figs. 4H, 5A and 5D). One pair of long setae in the last annules of the dorsal part of the tail (Figs. 4H, circle and 5D). Each setae with a collar at the base and close to each other (50.44 µm long). Ventrally, there is one pair of short setae with also a collar at the base (11–13 µm long). Two pairs of setae are present on the last annules of the tail, each setae featuring a collar at its base and positioned close to one another, measuring 52–56 µm in length (Fig. 5D). Additionally, there is another pair of shorter setae, also with collars at their bases, measuring 12–14 µm in length (Figs. 4G, 4H, 5D). Five more setae are distributed along the tail, in similar size as in males, in each somatic row of setae. One or two minute setae (1.5 µm long) are located at the non-annulated tail tip. In some specimens, one of these setae (and also for those on the annulated tail) is absent, though its minute insertion point remains visible (Fig. 4H). Non-annulated tail tips are long, constituting 56–59% of total tail length. Caudal glands not evident.

## Juveniles paratypes

**Juvenile third stage.** Body shape similar to adults. Body length 426.3 µm long, head diameter 19.3 µm long and a pharynx 75.3 µm long, with a minimally developed isthmus (Fig. 6A). Swollen region representing 24% of the total length. Amphideal fovea is smaller in both length (9.2 µm) and width (4 µm), circular and closed-shape (Fig. 6C). Several pedicel setae with 2.18 µm long, in some specimens as long as in some in adults (Fig. 6C). Two CATs in the dorsal region of the helmet, all at the level of the amphid (15.89–22.24 µm long) (Fig. 6C). Five PATs in both subventral and sublateral row (Figs. 6A, 6D). Tail slender, with the non-annulated tail tip corresponding to 50.7% of the tail length, similar to that in adults (Fig. 6D). The cuticle of the non-annulated tail tip end ornamented with minute punctations. All observed juveniles exhibit a globular appearance on the lateral sides of the body between the body wall and the cuticle, sometimes with brownish or yellowish coloration in the pharyngeal and anterior region.

**Juvenile fourth stage.** Body shape similar to adults. Body length 514.3 µm long, with head diameter 24.3 µm long. Swollen region representing 25.2% of the total length (Fig. 6B). Amphideal fovea 80.6 µm, circular and closed, similar to the third stage. Presence of pedicel setae with 2.82 µm long. Three CATs located on the helmet with the longer one and

more evident with 21.2 μm long. Seven PATs on both sublateral and subventral row, first SlAT with 50.4 μm long and first SvAT with 44.3 μm long (Fig. 6B).

**Diagnosis.**

*D. miguelitus* **sp. nov.** is characterized by the presence of four CATs located in the dorsal side of the helmet at the level of the amphid. All CATs situated anterior to the cuticular annulations. Amphid elongate loop-shaped with different branch sizes in males and females and circular in juveniles. A collar is present at the base of some setae, as a pedicel-like structures. Paravulvar setae are present in two pairs, one anterior and one posterior to the vulvar aperture. Additionally, a minute setae is visible parallel to the vulvar aperture. In the males, PATs are arranged in four longitudinal rows: two sublateral rows each with 10–12 CATs, and two subventral rows consisting of 10 PATs each one. In the females, sublateral and subventral rows with 13 CATs each one. Two pairs of setae in the annulated part of the tail, one pair with long setae (40–50 μm and one pair with shorter setae (11–13 μm). The non-annulated tail tip corresponds to 40–44% in males, 56–59% in females, and 50–55% in juveniles.

## Differential diagnosis and relationship

*Dracograllus miguelitus* **sp. nov.** is immediately distinguished from its congeners by possessing only four cephalic adhesive tubes (CATs) located at the level of the amphid, in contrast to species with six CATs (*D. minutus*), eight CATs (*D. antillensis*, *D. chitwoodi*, *D. cobbi*, *D. demani*, *D. eira*, *D. filipjevi*, *D. gilbertae*, *D. grootaerti*, *D. kreisi*, *D. laingensis*, *D. mawsoni*, *D. ngakei*, *D. papuensis*, *D. pusillus*, *D. solidus*, *D. spinosus*, *D. timmi*, *D. trispinosum*, *D. trukensis*, and *D. wieseri*), or more, such as *D. cornutus* (11 CATs), *D. falcatus* (12 CATs), *D. gerlachi* (13 CATs), and *D. stekhoveni* (14 CATs).

The absence of cuticular ornamentation further differentiates *D. miguelitus* **sp. nov.** from species with spines (*D. antillensis*, *D. chitwoodi*, *D. grootaerti*, *D. minutus*, and *D. trukensis*) or dot-like punctations (*D. filipjevi*, *D. gerlachi*, *D. kreisi*, *D. pusillus*, and *D. trispinosum*). The cuticle of *D. miguelitus* **sp. nov.**, with the collars at the bases of its setae, resembles those observed in *D. cobbi*, *D. mawsoni*, *D. filipjevi*, and *D. timmi*, though the setae in these species are significantly smaller compared to those in *D. miguelitus*. The presence of paravulvar setae distinguishes *D. miguelitus* **sp. nov.** from several species, including *D. chitwoodi*, *D. cobbi*, *D. cornutus*, *D. filipjevi*, *D. grootaerti*, *D. minutus*, *D. ngakei*, *D. pusillus*, *D. solidus*, *D. spinosus*, *D. stekhoveni*, *D. timmi*, *D. trispinosum*, and *D. trukensis*, all of them lacking setae at the vulva.

*Dracograllus miguelitus* **sp. nov.** is geographically closest to *D. demani* and *D. trispinosum* but can be distinguished from these species by several morphological features. See Tables 2 and 3 for the morphometrical and descriptive comparison between all the valid species In addition to the number of cephalic adhesive tubes (CATs)—four in *D. miguelitus* **sp. nov.** compared to eight in both *D. demani* and *D. trispinosum*—the new species differs in the number of sublateral adhesive tubes (10–12 in *D. miguelitus* **sp. nov.** *vs* six in *D. demani* and 10 in *D. trispinosum*), subventral adhesive tubes (10 in both *D. miguelitus* **sp. nov.** and

**Table 2 Morphometrical comparison for all valid species of *Dracograllus* genus including *Dracograllus miguelitus* sp. nov.**

| Species | L | CATn | SlATn | SvATn | Spicule | Non-annulated tail tip % |
|---|---|---|---|---|---|---|
| *D. antillensis Decraemer & Gourbault, 1986* | Ms: 410–510 | 8 | Ms: 6–10 | Ms: 9–14 | 36–40 | Ms:14–20 |
| | Fs: 410–510 | | Fs: 7–9 | Fs: 8–12 | | Fs: 29–32 |
| *D. chitwoodi Allen & Noffsinger, 1978* | Fs:500–600 | 8 | Fs: 9–10 | Fs: 8–10 | – | Fs: 50–54 |
| *D. cobbi Allen & Noffsinger, 1978* | M:500 | 8 | M: 9 | M: 12 | 51 | M: 44 |
| | F: 500 | | F: 8 | F: 14 | | F: 53 |
| *D. cornutus Decraemer, 1988* | Ms: 495–610 | 10–11 | Ms: 16 | Ms: 13 | 55–56 | Ms: 26–28 |
| | Ms: 480 | | Ms: 18 | Ms: 16 | | Ms: 49 |
| *D. demani Allen & Noffsinger, 1978* | Ms: 500–800 | 8 | Ms: 5–7 | Ms: 8–12 | 45–53 | Ms: 24–39 |
| | Fs: 500–800 | | Fs: 6–8 | Fs: 10–13 | | Fs: 41–51 |
| *D. eira* (*Inglis, 1968*) | M: 500 | 8 | M: 12 | M: 8 | 48 | M: 48 |
| | F: 600 | | Fs: 12 | Fs: 8 | | F: 41 |
| *D. falcatus* (*Irwin-Smith, 1918*) | M: 800 | 12 | M: 12 | M: 17 | 71 | M: 32 |
| | F: 900 | | F: 21 | F: 23 | | F: 48 |
| *D. filipjevi Allen & Noffsinger, 1978* | Ms: 500–700 | 8 | Ms: 8–11 | Ms: 9–11 | 37–40 | Ms: 40–50 |
| | Fs: 600–700 | | Fs: 12–14 | Fs: 9–11 | | Fs: 46–55 |
| *D. gerlachi Allen & Noffsinger, 1978* | M: 600 | 13 | M: 13 | M: 18 | 39 | M: 28 |
| | F: 700 | | F: 24 | F: 21 | | F: 28 |
| *D. gilbertae Verschelde & Vincx, 1993* | M: 581 | 8 | M: 10 | M: 9 | 59 | M: 20 |
| | F: 639 | | F: 13 | F: 10 | | F: 42 |
| *D. grootarti Decraemer, 1988* | M: 650 | 8 | M: 10 | M: 14 | 68 | M: 46 |
| | Fs: 675–755 | | Fs: 12–13 | Fs: 7–9 | | Fs: 43–61 |
| *D. kreisi Allen & Noffsinger, 1978* | M: 400 | 8 | M: 5 | M: 11 | 36 | M: 40 |
| | F: 400 | | F: 12 | F: 9 | | F: 69 |
| *D. laingensis Decraemer, 1988* | M: 460 | 8 | M: 8–9 | M: 8 | 39 | M: 24 |
| | F: 440 | | F: 5 | F: 5 | | F: 43 |
| *D. mawsoni Allen & Noffsinger, 1978* | Ms: 500–600 | 8 | Ms: 13 | Ms: 13 | 52–54 | Ms: 28–35 |
| | F: 700 | | F: 15 | F: 16 | | F: 58 |
| *D.miguelitus* sp. nov. | Ms: 630–735 | 4 | Ms: 10–12 | Ms: 10 | 47–54 | Ms: 40–41 |
| | Fs: 748–788 | | Fs: 13 | Fs: 13 | | Fs: 56–59 |
| *D. minutus Decraemer, 1988* | M: 290 | 6 | M: 5 | M: 2–3 | 18 | M: 24 |
| *D. ngakei Leduc & Zhao, 2016* | M: 576 | 8 | M: 11 | M: 10 | 50 | M: 28 |
| | Fs: 586–615 | | Fs: 13 | Fs: 10–12 | | Fs: 37–50 |
| *D. papuensis Decraemer, 1988* | M: 310 | 8 | M: 10 | M: 11 | 29 | M: 75 |
| | Ms: 350–400 | | Ms: 9–11 | Ms: 9–11 | | Ms: 46–56 |
| *D. pusillus Decraemer, 1988* | M: 310 | 8 | M: 10 | M: 6 | 26 | M: 28% |
| *D. solidus* (*Gerlach, 1952*) | M: 700 | 8 | M: 7 | M: 11 | 46 | M: 29 |
| | Fs: 600–800 | | Fs: 8–11 | Fs: 8–11 | | Fs: 46 |
| *D. spinosus Decraemer, 1988* | M: 340 | 8 | M: 8 | M: 10 | 45 | M: 49% |
| *D. stekhoveni Allen & Noffsinger, 1978* | Ms: 500–600 | 14 | Ms: 16–23 | Ms: 16–23 | 40–50 | Ms: 22–34 |
| | Fs: 500–600 | | Fs: 20–25 | Fs: 21–29 | | Fs: 37–47 |

| Table 2 (continued) | | | | | | |
|---|---|---|---|---|---|---|
| Species | L | CATn | SlATn | SvATn | Spicule | Non-annulated tail tip % |
| *D. timmi* Allen & Noffsinger, 1978 | Ms: 500–700 | 8 | Ms: 7–10 | Ms: 19–23 | 41–51 | Ms: 29–36 |
|  | Fs: 500–600 | | Fs: 9–12 | Fs: 7–11 | | Fs: 43–52 |
| *D. trispinosus* (Allen & Noffsinger, 1978) | Ms: 700 | 8 | Ms: 10 | Ms: 6–7 | 59–64 | Ms: 26–27 |
|  | Fs: 600–800 | | Fs: 12–13 | Fs: 8–13 | | Fs: 52–61 |
| *D. trukensis* Min et al., 2016 | Ms: 593–642 | 8 | Ms: 10 | Ms: 8–10 | 34–42 | Ms: 43–48 |
|  | Fs: 663–778 | | Fs: 13–15 | Fs: 9–11 | | Fs: 45–58 |
| *D. wieseri* Allen & Noffsinger, 1978 | M: 600 | 8 | M: 17 | M: 13 | 46 | M: 26 |
|  | F: 500 | | F: 14 | F: 12 | | F: 45 |

**Note:**
L, body length; CATn, number of cephalic adhesive tubes; SlATn, number of sublateral adhesive tubes; SvATn, number of subventral adhesive tubes; Ms, type series males; M, male holotype; Fs, type series females; F, female holotype. L and spicule measurements expressed in μm.

*D. demani*, but seven in *D. trispinosum*), and spicule length (50 μm in *D. miguelitus* sp. nov., compared to 37 μm in *D. demani* and 61 μm in *D. trispinosum*). Additionally, the non-annulated tail tip of *D. miguelitus* **sp. nov.** is longer (44% of body length in the male holotype and 56% in the female paratype) compared to the shorter tail tips in *D. demani* and *D. trispinosum* (32% and 26%, respectively). Morphometrical and descriptive comparison between all the valid species (Tables 2 and 3).

## Comments on the imaging approach

Several challenges related to the study of marine nematodes have been discussed here, and we would like to emphasize one of the most important ones: the difficulty in observing and measuring their morphological structures. Therefore, it is crucial to investigate advanced imaging methods to facilitate identification and capture additional morphological features (*Foulon, Malloci & Zeppilli, 2025* in press). The pioneering work of *Zullini & Villa (2009)* first documented the autofluorescence using confocal microscopy of free-living nematodes, with the redescription of *Eutobrilus andrassyi*. Other examples include the redescriptions of *Craspodema reflectans* (Cyatholaimidae) and *Longicyatholaimus maldivarum* (Cyatholaimidae) by *Semprucci & Burattini (2015)* and *Semprucci et al. (2017)*, respectively. Additionally, an introduction to the application of confocal techniques for observing marine nematodes is provided in *Semprucci et al. (2016)*. In our study, 3D fluorescence imaging has proven effective in several key aspects of identification, particularly for the Draconematidae family. We observed that various structures, that are difficult to study with traditional microscopic techniques, were analyzed with relative ease in our study. These included the insertion of the CATs, the cuticular ornamentation or annulations, and the number of rows of setae. Videos of the 3D fluorescence captures and additional pictures are available in the Supplemental Material (S1–S4).

## Biodiversity, distribution and ecology

In terms of species diversity, *Dracograllus* is the largest genus in the Draconematidae family, with 26 valid species (*Nemys, 2024*). These species are distributed across a variety of marine environments, and despite their large distribution, the genus is predominantly

**Table 3 Descriptive comparison for all valid species of *Dracograllus* genus including *Dracograllus miguelitus* sp. nov.**

| Specie | Ann. Ornam. | Fov. M. | Fov. F. | PS | Anal flap | Diff. Diagnosis |
|---|---|---|---|---|---|---|
| *D. antillensis* | Spine-like | Large, conspicuously 'U'-shape with ventral arm often slightly longer than dorsal | – | – | Absent | Spicules 35–40 μm long, arcuated and cephalated. Gubernaculum 11–15 μm long, with *corpus* and lateral wind. Four long somatic setae between the eighteenth SlATs. |
| *D. chitwoodi* | Spine-like | – | Elongated loop-shape | Absent | Present. short | Fewer SlATs and SvATs, absence of PS, and setae without collar at the base. |
| *D. cobbi* | Without | Elongated loop-shape | Elongated loop-shape | One pair anterior to the vulva (6–7 μm long) | Absent | Great number of SlATs in males, shorter caudal glands and anterior position of the vulva. |
| *D. cornutus* | Without | Short loop-shape | Short loop-shape | Absent | Absent | Similar to Dracotoramonema *Allen & Noffsinger, 1978*, but cornifor setae and length of SlATs less conspicuous than in *Dracotoramonema trispinosum* |
| *D. demani* | Without | Elongated loop-shape | Elongated loop-shape | Two setae (7–9 μm long) | Absent | PS in ventro-sublateral rows, but only anterior to the SlATs. |
| *D. eira* | Without | Elongated loop-shape | Elongated loop-shape | Absent | Absent | All CATs anterior to the amphid, and 1 SlAT on the non-annulated tail region. Males with SlATs posterior to the anus. |
| *D. falcatum* | Without | Elongated loop-shape | Elongated loop-shape | Two pairs, one anterior and one posterior to the vulva | Absent | Rostrum without Ceph Acan-set and with 12 CATs |
| *D. filipjevi* | Dot-like | Elongated loop-shape | Elongated loop-shape | Two pairs, one anterior and one posterior to the vulva. (5–6 μm long) | Absent | Scattered minute spiny on cuticle, Absence of PS. |
| *D. gerlachi* | Dot-like punctations, more evident at mid-body | Elongated loop-shape | Elongated loop-shape | Two pairs, one anterior and one posterior | Absent | 13 CATs on rostrum and great swollen esophageal region |
| *D. gilbertae* | Broad interannual space, ornamented with a slit | Large, ventrally whorled, open loop-shape | Large, closed loop-shaped | Absent | Absent | Large amphideal fovea, long and slender PATs. Slender tail with ventral post cloacalpostcloacal swelling. Spicules long and well cuticularized gubernaculum. |
| *D. grootaerti* | Spine-like | Long, inverted U-shaped, with longer ventral arm extending to the first annule | As in male, but shorter | Absent | Absent | Long body, with spiny ornamentaded annulated cuticle. Two of the SlATs in females on the tail region. |

| Specie | Ann. Ornam. | Fov. M. | Fov. F. | PS | Anal flap | Diff. Diagnosis |
|---|---|---|---|---|---|---|
| *D. kreisi* | Dot-like punctations | Elongated loop-shape | Elongated loop-shape | Absent | Absent | Absence of PS in ventro-sublateral row and shorter spicules in males. |
| *D. laingensis* | Spine-like | Long, inverted U-shaped | Elongated unispiral | Absent | Absent | Long swollen pharyngeal regions, and stiff posteriorly directed setae anterior to PATs. |
| *D. mawsoni* | Without | Elongated loop-shape | Slightly smaller, with more open loop than in male | Two pairs, one anterior and one posterior to the vulva. (4–7 µm long) | Absent | Great number of SlATs in males, and females with 1 SlAT posterior to anus |
| *D. miguelitus* sp. nov. | Without | Elongated loop-shaped, ventrally coiled, ventral arm slightly longer | Inverted U-shaped with branches more equal in size and more closed than in males | Two pairs, one anterior and one posterior to the vulva (6 µm long). Single seta emerging from the vulvar aperture | Absent | 4 CATs on the rostrum, PS longer in males than in females. |
| *D. minutus* | Spine-like | Very large, loop-shape, ventrally whirled | – | – | Absent | Smaller body size within the genus, only six CATs on rostrum, short spicules. Largest fovea within the genus. |
| *D. ngakei* | Without | Loop-shaped, with two arms of equal length | Loop-shaped, with two arms of equal length | Absent | Absent | 11 SvATs per row in male, all anterior to anus. Females with 12 SvATs with one of themn posterior to anus. |
| *D. papuensis* | Finely annulated | Long, inverted U-shaped, ventrally coiled, ventral arm slightly longer | Large, loop-shaped. dorsal arm slightly longer than ventral one | Minute setae: two ventral posterior and one anterior to the vulva | Absent | Shorter swollen pharyngeal region, spicule and c-value. |
| *D. pusillus* | Dot-like punctations at ring edges in the pharyngeal region | Long, inverted U-shaped. ventrally coiled, ventral arm slightly longer | – | – | Absent | Short and stout body with minute spiny ornamentations, short spicule. Long non-annulated tail tip. |
| *D. solidus* | Without | Elongated loop-shape | Elongated unispiral | Absent | Absent | 11 long setae intermingled with SlATs in males, unispiral amphid and 2 SlATs posterior to anus in females. |
| *D. spinosus* | Without | Large, oblique loop-shape by position of sublateral CAT, ventrally whirled; Ventral arm slightly longer than dorsal arm | – | – | Absent | Spiny ornamentation at the insertion base of several somatic setae in the posterior body region. Females and juveniles not found. |

(Continued)

| Specie | Ann. Ornam. | Fov. M. | Fov. F. | PS | Anal flap | Diff. Diagnosis |
|---|---|---|---|---|---|---|
| *D. stekhoveni* | Without | Elongated loop-shape | Elongated loop-shape | Two pairs, one anterior and one posterior to the vulva. (3–5 µm long) | Absent | 1 pair of sub-lateral cephalic acant setae on rostrum. |
| *D. timmi* | Spine-like projections | Elongated loop-shape, some specimens ventral arm curved anteriorly toward dorsal arm almost forming unispiral | Elongated loop-shape | Two pairs, one anterior and one posterior to the vulva. (5–7 µm long) | Absent | Faint annular ridges with spine-like projections appearing as 2 rows of fine punctations. |
| *D. trispinosum* | Dot-like punctations | Very large, loop-shape | Elongated unispiral | – | Absent | Males with 3 large Corn-set, a single ventral mid-body setae and 1 preanal pair. |
| *D. trukensis* | Ridges with spiny protrusion, spiny ornamentation | Large. Elongated, open loop-shaped, longer ventral arm extending to the first body cuticular annule | Large, elongated and closed loop-shaped, shorter than in male | Absent | Absent | Numerous minute spiny ornamentation on male and female cuticle. Shorter spicule in males. |
| *D. wieseri* | Granules and vacuoles | Elongate loop-shape | Elongate loop-shape | Absent | Present. Short | 6 long setae intermingled with SlATs in males, and SlAT 1 in females posterior to anus. |

**Note:**
Ann. Ornam., annules ornamentation; Fov. M., amphideal fovea male; Fov., amphideal fovea female; Parav. set., paravalvular setae; "–": not provided in the original description or not applicable.

associated with shallow, tropical and subtropical regions (*Min et al., 2016*). The distribution and general ecological characteristics of all *Dracograllus* valid species are presented in Table 4. The *Dracograllus* genus includes species distributed across the Pacific (18 species), Atlantic (nine species) and Indian (four species) oceans. Some species, such as *D. eira*, occur in multiple oceanic regions, underscoring their adaptability to diverse oceanic regimes. Distribution of all valid species, and species occurrences including non-identified *Dracograllus* specimens are provided in Fig. 7. The Pacific Ocean is the region where the highest number of *Dracograllus* species have been both recorded and described, likely reflecting a bias due to a more extensive sampling. Examples *include D. cornutus, D. falcatus, D. filipjevi, D. gerlachi, D. grootaert, D. laingensis, D. mawsoni, D. minutus, D. papuensis, D. pusillus, D. spinosus, D. timmi, D. trukensis*, and *D. wieseri* from a variety of habitats and environmental conditions, particularly in coastal regions. More recently, *Leduc & Zhao (2016)* described *D. ngakei*, a species from intertidal coarse sand and gravel sediments in New Zealand, including molecular and morphological data.

In the Atlantic Ocean, species such as *D. antillensis, D. chitwoodi*, and *D. kreisi* are typically found in shallow marine environments, often associated with sandy beaches on intertidal or subtidal zones. *Decraemer & Gourbault (1986)* found approximately 500 individuals of *D. antillensis* in samples from Guadeloupe, a notably high number for a single species, especially when compared to the abundances typically observed in

Table 4 **Distribution and ecological characteristics of *Dracograllus* species.**

| Species/reference | Ocean | Geographic distribution | Habitat | Habitat type, sampling and conditions | Remarks |
|---|---|---|---|---|---|
| *D. antillensis Decraemer & Gourbault, 1986*\*; *Stock & Nadler, 1998* | Atlantic | Guadeloupe Island: Anse de la Gourde, Grande Terre; Les Galets, Capesterre; Petite Anse, La Marie-Galante. Martinique Island: Anse l'Étang; Anse Figuiers. | Intertidal region | Sandy beach; interstitial waters with coarse and calcareous sediments. | – |
| *D. chitwoodi Allen & Noffsinger, 1978*\* | Atlantic | Coral Key, Florida, USA | Subtidal region | Sandy beach; sediment associated with calcareous algae (*Halimeda sp.*). | Males only measured, without complete description. No third or fourth-stage juvenile observed. |
| *D. cobbi Allen & Noffsinger, 1978*\*; *Decraemer, 1988*. | Atlantic | Coral Key, Florida, USA; Anse de la Gourpe, Guadeloupe. | Intertidal region | Sandy beach; sediment associated with calcareous algae (*Halimeda sp.*). | Females from Guadeloupe lack paravalvular setae and show other differences compared to the original description. See *Decraemer, 1988*. |
| *D. cornutus Decraemer, 1988*\* | Pacific | Laing Island, Papua New Guinea and River Mouth, NT, Australia | Subtidal region | Sandy beach; sediment sampling | - |
| *D. demani Allen & Noffsinger, 1978*\*; *Decraemer, 1988*; *Verschelde & Vincx, 1993*; *Shahina et al., 2019* | Atlantic, Pacific and Indian | Marseille, France; Laing Island, Duangit Reef, Papua New Guinea; Malindi, Kenya; Pakistan. | Subtidal region, down to 42 m depth | Sandy beach; coarse sand with algae and coarse coral sand | Specimens from Papua New Guinea differ from the type locality by having a shorter general body length, shorter PATs, and shorter spicules. |
| *D. eira Inglis, 1968*\*; *Decraemer, 1988*; *Verschelde & Vincx, 1993* | Pacific and Indian | St. Vincent's Bay, New Caledonia; Laing Island, Papua New Guinea; Malindi, Kenya | Subtidal and intertidal zone | Sediments associated with polychaete tubes and large pieces of dead coral. | – |
| *D. falcatus Irwin-Smith, 1918*\*; *Allen & Noffsinger, 1978* | Pacific | Cremorne, Port Jackson, New South Wales, Australia; Long Reef and Vaucluse, Australia | Subtidal region, from 1.2 - 1.5 m depth | Sandy beach; sediment sampling with seaweed and shells | – |
| *D. filipjevi Allen & Noffsinger, 1978*\*; *Rho, Kim & Kim, 2006* | Pacific | Oarai, Ibaraki-ken, Honshu Island, Japan; Daebo-ri, Guryongpo, Korea | Subtidal region | Washings of holdfasts of Kelps and also in shallow littoral calcareous algae | – |
| *D. gerlachi Allen & Noffsinger, 1978*\*; *de Jesús-Navarrete* | Atlantic and Pacific | Ibusuki, Kyushu Island, Japan and Laguna de Términos, Gulf of Mexico | Subtidal region | Sandy beach; sediment sampling with brown algae growing on rocks | – |
| *D. gilbertae Verschelde & Vincx, 1993*\* | Indian | Gazi, Kenya | Subtidal region | Sandy beach; core of 3.5 cm diameter into the sediment down to 20 cm depth, close to mangrove plants (*Sonneratia* sp.) | – |
| *D. grootaerti Decraemer, 1988*\* | Pacific | Madang Province, Hansa Bay, Duangit Reef, Laing Island, Papua New Guinea | Subtidal region at 42 m depth. | Sandy beach; sediment sampling with polychaete tubes, and coral sand | – |

*(Continued)*

| Species/reference | Ocean | Geographic distribution | Habitat | Habitat type, sampling and conditions | Remarks |
|---|---|---|---|---|---|
| *D. kreisi Allen & Noffsinger, 1978** | Atlantic | Coco Solo, on Galeta Beach, Panama | Subtidal region | Sediment associated with calcareous algae (*Halimeda sp.*). | – |
| *D. laingensis Decraemer, 1988** | Pacific | Laing Island, Papua New Guinea | Subtidal region at 42 m depth. | Sediment sampling with polychaete tubes, and coral sand | – |
| *D. mawsoni Allen & Noffsinger, 1978** | Pacific | Long Nose Point, Port Jackson, New South Wales, Australia | Subtidal region | Sandy beach; sediment sampling with bottom debris | – |
| *D. minutus Decraemer, 1988** | Pacific | Laing Island, Papua New Guinea | Subtidal region | Sediment sampling with polychaetes tubes of sand and mucus | No female or juvenile known |
| *D. miguelitus* sp. nov. Johnson et al., 2024* | Atlantic | Lucky Strike vent field, Mid Atlantic Ridge | Deep-sea | Hydrothermal inactive vent structure; | Only present in the inactive vent structure at LS, without individuals in active or periphery samples. |
| *D. ngakei Leduc & Zhao, 2016** | Pacific | Hataitai Beach, Wellington, New Zealand | Mid-intertidal region | Sandy beach; sediment sampling (0 to 10 cm sediment depth) with coarse sand and gravel | SSU Molecular sequences available in original description. |
| *D. papuensis Decraemer, 1988** | Pacific | Laing Island, Papua New Guinea | Subtidal region | Sediment sampling with dead coral debris, also with polychaete tubes of sand and mucus | Only one male found, without non-annulated tail tip length known. |
| *D. pusillus Decraemer, 1988** | Pacific | Laing Island, Papua New Guinea | Subtidal region | Sediment sampling, with dead coral debris | – |
| *D. solidus Gerlach, 1952** | Atlantic and Indian | Banyuls, France; Bay of Biscay, Mediterranean sea; Mascarene Islands. | Subtidal region | Sandy beach; sediment sampling | Also recorded in Mascarene Islands, no juvenile known. |
| *D. spinosus Decraemer, 1988** | Pacific | Laing Island, Papua New Guinea | Subtidal region | Sediment sampling with polychaete tubes, sand and mucus | Males and juveniles not found. |
| *D. stekhoveni* Allen & Noffsinger, 1988* | Pacific | Solano, Colombia; Port Jackson, Australia; Isla Taboga, Panama | Subtidal region | Sandy beach; sediment sampling with corals | Juveniles specimens third stage without PS. |
| *D. timmi Allen & Noffsinger, 1978** | Pacific | Bora Bora Island, Society Islands | Subtidal region | Sandy beach; sediment sampling of coarse sand | Second and third-stage juveniles not found |
| *D. trispinosus Allen & Noffsinger, 1978** | Atlantic | Southwest of the Pomegues Ratonneau jetty, near Marseille, France | Subtidal region | Sandy beach; sediment sampling at 20 m depth. | – |
| *D. trukensis Min et al., 2016** | Pacific | Weno, Chuuk, Micronesia | Subtidal region | Sandy beach; sediment sampling with seagrass bed (*Zostera* sp., from 1 to 2 m depth) | – |
| *D. wieseri Allen & Noffsinger, 1978** | Pacific | Juan Fernandez Islands, Chile | Subtidal region during high tide zone | Sandy beach; sediment sampling with green algae | – |

**Note:**
References marked with an asterisk (*) indicate the original description and type locality, while those without an asterisk refer to additional localities.

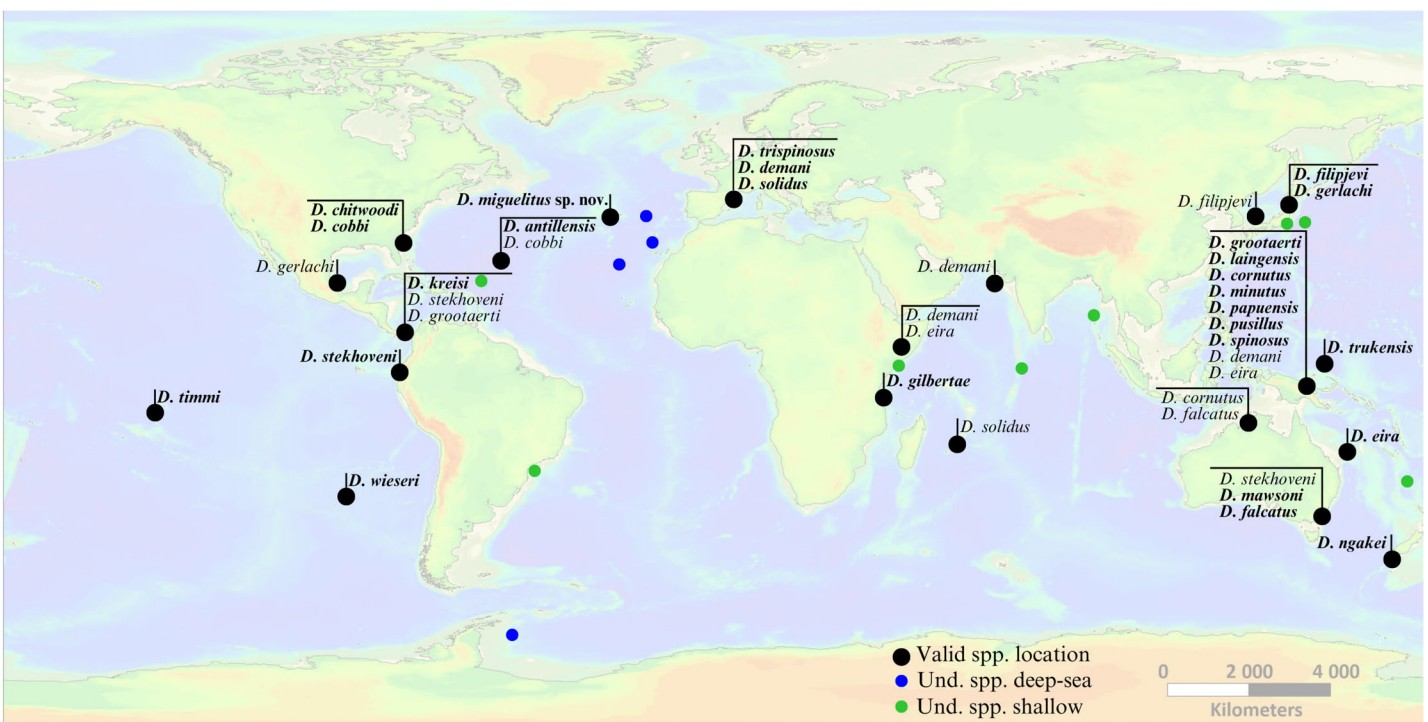

**Figure 7 Global occurrence locations of the genus *Dracograllus*.** Black dots indicate type localities of valid species (holotype names in bold, non-bold for paratypes). Colored circles represent occurrence locations of unidentified individuals or those classified as morphotypes (green points indicate these occurrences in shallow waters, 0–200 m, and blue points in deep-sea habitats, >200 m). Horizontal lines group morphotypes with overlapping occurrences or geographically close localities (*e.g.*, all species described for the Papua New Guinea region). For the precise locations, habitats, sampling details, and remarks on each valid and undetermined species globally, refer to Tables 4 and 5, respectively. *Nomen nudum* and invalid species in general not shown, but available in the genus review section.

*Dracograllus* and even within the broader Draconematidae family. *Allen & Noffsinger (1978)* described *D. trispinosus* at 20 m depth, revealing the species occurrence in subtidal zones.

The Indian Ocean hosts fewer described species, yet their habitats share similarities with those in other regions. For instance, *D. demani* has been reported in tidal coral sands along the Malindi coast and also, in similar sandy habitats in the Pacific Ocean. Likewise, *D. eira* is known from both the Atlantic Ocean, where it inhabits subtidal sandy zones, and the Indian ocean, where it has been recorded in mangrove-associated sediments, demonstrating its ability to thrive in a range of coastal habitats. *D. solidus*, another widespread species, has been documented in the Atlantic, Pacific, and Indian oceans, consistently associated with coarse sand in subtidal regions.

Only a few studies have investigated the spatial distribution of *Dracograllus* in deep-sea ecosystems. Four morphotypes were found at the summit of the GSM in the Atlantic, which is a flat plateau covering more than 1,400 km$^2$, with 293 to 511 m depth (*Pfannkuche, Sommer & Kähler, 2000*). They exhibit significant abundance compared to other Draconematidae genus, with non-overlapping occurrences between each morphotype. The specific sedimentary processes on the GSM (*Levin & Nittrouer, 1987*), combined with the erosion of old coral reefs, create a coarse sedimentary environment,

with small and morphologically complex biogenic structures covering the substratum (*Pasenau, 1971*; *Nellen, 1998*). This wide variety of ecological niches would explain their high abundance and the co-occurrence of several species. Similarly, *Zeppilli et al. (2014)* reported several *Dracograllus* specimens at the Condor Seamount (CS), at 206 m depth, in the Azores archipelago (Northern Atlantic). The summit of this structure exhibited a highest species richness and dominance for several genus, clearly differing from the surrounding deep-sea habitats or along other CS habitats. The summit was the only area of the study where *Dracograllus* sp. 1 was present, a flat region, covered by biogenic structures such as sponge sediments or corals, as observed for most *Dracograllus* species. Similar results regarding Draconematidae species were obtained on other biogenic and sedimentary habitats (*Willems et al., 1982*; *Ndaro & O'lafsson, 1999*; *Raes & Vanreusel, 2006*; *Raes et al., 2007*; *Raes, Decraemer & Vanreusel, 2008*).

Draconematidae species were recently observed in samples collected from two deep hydrothermal vent fields, TAG and Snake Pit (SP) (*Spedicato et al., 2020*) located on the Mid-Atlantic Ridge. *Dracograllus* sp. was present in 50% of the SP samples, occurring in reddish sediments covered by polychaete tubes. In contrast, these features were absent or less evident at TAG, where only *Cephalochaetosoma* was recorded. Environmental conditions differed significantly between the vent fields. The total sulfur content in the sediment profiles (0–5 cm) was higher at SP than at TAG and the oxygen penetration about ten times lower at SP. High concentrations of sulfur can lead to death due to the inhibitory action of $H_2S$ on cytochrome c oxidase, an essential enzyme for aerobic respiration. This mechanism blocks the electron transport chain, disrupting ATP production and resulting in metabolic collapse (*Bagarinao, 1992*). However, some nematode species have developed strategies to cope with sulfide toxicity, such as the oxidation of $H_2S$ into elemental sulfur and its deposition in the epidermis, a process observed in *Oncholaimus campylocercoides* (*Thiermann, Vismann & Giere, 2000*). This ability to accumulate and later remove elemental sulfur may enable nematodes to colonize sulfide-rich environments, exploiting niches where most organisms cannot survive. Moreover, body elongation and a higher surface-to-volume ratio may help them cope with low $O_2$ levels in habitats with limited oxygen availability (*Vanreusel et al., 2010b*).

The non-overlapping distribution of *Dracograllus* species at deeper sites suggests that each species may have specific habitat requirements, influenced by both the nature and composition of the substratum, as well as the level of hydrothermal activity. The type-habitat of *D. miguelitus* **sp. nov.**, is characterized by low hydrothermal influence compared to active sites at the Lucky Strike (LS) vent field (*Chavagnac et al., 2018*). However, residual venting activity is still present, evidenced by the presence of manganese oxide-hydroxide and high CH4 concentrations measured above the substratum in one of the samples. Environmental conditions, including sediments rich in sulfide minerals, can stimulate microbial communities, which are essential as primary producers in these deposits (*Van Gaever et al., 2009*). This creates a higher food resource availability and provides structural conditions suitable for the occurrence of Draconematidae species, including *D. miguelitus* **sp. nov.**, classified as microbial feeders. In summary, the residual hydrothermal activity likely promotes microbial growth, a significant food source for

bacterivores and microbial feeders like *D. miguelitus* **sp. nov**. and most Draconematidae species. Moreover, the presence of biogenic structures like microbial mats, appear to play a role in the distribution of *Dracograllus* species. These species are capable of using adhesive tubes to anchor to these structures and may also feed on them (*Raes et al., 2007*).

Prior to the description of *D. miguelitus* **sp. nov.**, the Snake Pit species were the deepest-known representatives of the genus, found at depths between 3,480–3,570 m.

## Biogeography and evolutionary perspectives

Several species of *Dracograllus*, and Draconematidae in general, have been found in only a few locations beyond their type habitats. However, nematologists agree that cosmopolitanism is common among various species and groups of marine nematodes (*Decraemer, Gourbault & Helléouet, 2001*). As reflected by *Gad (2009)*, based on Draconematidae species, one important starting point to determine the origins of these species is to identify their closest relatives and where they occur. In fact, some of the closely related species of *Dracograllus miguelitus* **sp. nov.** (*D. demani* and *D. minutus*) inhabit coastal and sublittoral environments of the Mediterranean Sea. These Mediterranean regions could be the source of this species, as surface currents transport waters from Gibraltar toward the Azores and upper regions of the North Atlantic (*Dietrich et al., 1975*; *Brenke, 2002*). Drifting-buoy experiments have confirmed that Mediterranean water eddies travel westward from Gibraltar across the Atlantic (*Richardson, 1996*). Such westward flows also occur at approximately 900 m depth, facilitating the transport of fauna, including meiofauna, which may drift as eggs, juveniles, or adults attached to marine snow (*Pingree, García-Soto & Sinha, 1999*; *Gad & Schminke, 2004*). This may also be the case for some *Prochaetosoma* species, as congeners have also been recorded in the Mediterranean. Other potential source regions, such as the coasts of Mauritania or Morocco, remain unconfirmed due to the absence of Draconematidae records from these areas. If such transport occurred, it would also depend on Mediterranean water flows (*Gad, 2009*).

Studies by *Gad (2009)* on the Great Meteor Seamount, *Zeppilli et al. (2014)* on the Condor Seamount, and *Spedicato et al. (2020)* on the TAG and Snake Pit vent fields identified closely related species in various deep-sea habitats along the northern MAR. Fifteen Draconematidae species, spanning several genera—including *Draconema, Paradraconema, Eudraconema, Prochaetosoma, Cephalochaetosoma*, and *Tenuidraconema* —were recorded on the GMS plateau, located 500 km from the Lucky Strike vent field. Remarkably, fourteen of these species were new to science, including four *Dracograllus* morphotypes (Table 5). In the CS (~300 km from LS), a rich and exclusive nematode community was documented, with 35 species distributed across genera such as *Akanthepsilonema, Apenodraconema, Bathychaetosoma, Dracograllus*, and others. Similarly, *Spedicato et al. (2020)* observed several Draconematidae specimens from three genera: *Cephalochaetosoma, Dinetia*, and *Dracograllus*.

Both hydrothermal vents and seamounts can be considered true oases of life compared to the surrounding deep-sea environment (*McClain et al., 2012*), emphasizing their importance for biogeographic studies, particularly for taxa that exhibit some degree of
**Table 5  Global distribution, habitats, and environmental characteristics of unidentified *Dracograllus* sp.**

| Ocean | Site | Coords. | Habitat | Substratum type | Morphospecies | Reference |
|---|---|---|---|---|---|---|
| Atlantic ocean | Great Meteor Seamount | 30°00′N, 28°30′W. | Plateau of the seamount, with 1,465 km2 | Calcareous biogenic sands | *Dracograllus sp.4*; *Dracograllus sp.5*; *Dracograllus sp.6*; *Dracograllus sp.7* | *Gad (2009)* |
| | Condor Seamount | 38°32.949′N, 29°02.879′W. | Summit of the seamount, at 206 m depth. | Large rocky outcrops, gravels, and coarse bioclastic deposits | *Dracograllus* sp. | *Zeppilli et al (2014)* |
| | Snake Pit vent field | 23°22.0′N, 44°57.0′W. | Sampling located 70 m from one black smoker. Depth between 3.480 m and 3.570 m | Reddish sediments covering the corer, with several polychaete tubes and individuals | *Dracograllus* sp. | *Spedicato et al. (2020)* |
| | Anse Laborde, Guadeloupe island | 16°29.2′N, 61°30.3′W | Intertidal beach zone with high hydrodynamic activity. | Composed of detrital fragments, mostly carbonates | *Dracograllus* sp. 1 | *Decraemer & Gourbault (1986), Renaud-Mornant & Gourbault (1981)* |
| | Raisins, clairs, Guadeloupe | 16.24892°N, 61.28345°W | Sandy beach on the characterized by a low sandy ridge (2 to 3 m in height) facing frequent waves and subject to significant coastal erosion | Sediments consist of a low sandy ridge, with a "beach-rock" (sandstone) layer along the coastline. | *Dracograllus* sp. 2 and *Dracograllus* sp. 3 | *Decraemer & Gourbault (1986), Renaud-Mornant & Gourbault (1981)* |
| | La Marie Galante. | 15°55′59.99″N, 61°15′60.00″W | Sandy Beach with a topography that includes a low sand ridge parallel to the shoreline and sparse vegetation | Sandy beach composed of sediments ranging from fine volcanic sands to coarse organogenic sands. | *Dracograllus* sp. 4 | *Decraemer & Gourbault (1986), Renaud-Mornant & Gourbault (1981)* |
| | Guadeloupe | 15.912°N, 61.269°W | | | | |
| | Guanabara Bay, Rio de Janeiro, Brazil. | 22°24′ S–22°57′S, 42°33′ W–43°19′W | Sandy Beach, intertidal zone. | Substratum composed of sand, from medium to very coarse sediments. Highly impacted beach region subject to anthropogenic pressures. | *Dracograllus* sp. | *Maria et al. (2008)* |
| Pacific ocean | Munseon Island, Jjeudo, Korea | 33°13′66″N, 126°34′18″E | Subtidal zone, 37 m deep. | Sampling based on washings of shallow subtidal detritus and coarse sediments | *Dracograllus* sp. 1 | *Rho & Min (2011)* |
| | Geomundo Island, Jeonranamd, Korea | 34°05′57″N, 127°14′84″E | Intertidal zone, associated with invertebrates | Substratum with associated invertebrates | *Dracograllus* sp. 2 | *Rho & Min (2011)* |
| | Volcanic Island of Moorea, French Polynesia | (17°30′ S–149°50′W) | Flat beaches surrounded by a large coral reef | Sediments with coarse coral sand | *Dracograllus* sp. 1 e *Dracograllus* sp. 2 | *Gourbault, Warwick & Helléouet (1995)* |

| Ocean | Site | Coords. | Habitat | Substratum type | Morphospecies | Reference |
|---|---|---|---|---|---|---|
| Indian ocean | Gazi Kenya | −4.4222°S, 39.5050°E | Sandy beach, intertidal zone | Sample taken in mangrove region, with *Ceriops sp.* tree | *Dracograllus* spec. | *Verschelde & Vincx (1993)* |
| | Chidiyatapu, South Andaman Island, India | 11°29'30"N–11°30'34"N, 92°35'10"E–92°42'30"E | Rocky coastal area | Sediments associated with several seagrasses patches (*Thalassia hempirichi*, *Halodule uninervis* and *Halophila ovalis*) | *Dracograllus sp.* | *Naufal & Padmavati (2018)* |
| | Marina Park, Andaman Islands | 11°40'15.39"N, 92°45'39.16"E | Sublitoral sediments | Substratum composed of silty-sand and clayey-sand | *Dracograllus sp.* | *Arunima et al. (2023)* |
| | Huvadhoo Atoll, Maldives | 08°33'20.88"N, 73°81'4.76"E | Central atoll region | Sediments with coarse and gravelly sand, at 61 m deep. | *Dracograllus sp.* | *Semprucci et al. (2014)* |
| Southern ocean | Halley Bay, Weddel Sea | 74°S–75°S, 25°W–29°W | Shelf break region, 500 m deep | Sediment poorly to extremely poorly sorted, with significant variations in grain size, with presence of pellite and gravel | *Dracograllus sp.* | *Vanhove, Arntz & Vincx, 1999* |

habitat exclusivity, as observed in Draconematidae in the North Atlantic. Another intriguing aspect of Draconematidae in these regions is their morphological variability, which may reflect underlying biogeographic processes (*Costello & Chaudhary, 2017*). For example, in *Dracograllus* species from the GMS, individuals from the southern part of the plateau possess a fully divided cephalic capsule (helmet), whereas those from the northern part have a partially divided one (*Gad, 2009*). Additionally, there are variations in the number of SlATs and SvATs. Several other distinctive traits were reported, including the presence of eight strong spines around the vulva in *Draconema* sp. 1, a long and conical cephalic capsule in *Cephalochaetosoma* sp. 10, and extra-wide annules in the pharyngeal region of *Prochaetosoma* sp. 12. None of these distinctive traits were observed in Draconematidae species from LS (*Tchesunov, 2015*; W Johnson, 2025, unpublished data).

The intrageneric variation in the helmet among *Dracograllus*, along with the non-overlapping distributions of several Draconematidae genera and species across the CS, GMS, Snake Pit, and LS, and this may be related to an ongoing speciation process (*George, 2004*; *Gad, 2004*; *Gad, 2009*), similar to what was observed by *George & Schminke (2002)* and *Gad & Schminke (2004)* in copepods and macrofaunal species, respectively. In fact, when closely related species exhibit significant morphological variations within small geographic regions, it suggests that species may be arising through micro-allopatric speciation, where populations diverge due to localized environmental differences, leading to subtle, but sometimes crucialmorphological distinctions (*Rundle & Nosil, 2005*). As these populations adapt to specific ecological niches, genetic divergence and reproductive isolation may drive the emergence of new species, highlighting the importance of understanding local biodiversity and the environmental factors influencing species differentiation.

Given the known limitations of morphology-based taxonomy, such as cryptic diversity and convergent evolution, future studies integrating molecular markers, such as COI or 18S, will be crucial for validating the observed patterns and refining our understanding of species connectivity and dispersal (*Palmer, 1988a*, *1988b*; *De Ley et al., 2005*; *Bhadury et al., 2006*; *Derycke et al., 2010*; *Curini-Galletti et al., 2012*; *Ahmed et al., 2015*; *Martínez García et al., 2023*). Despite these challenges, our findings, together with the limited existing data on species distributions, suggest that both oceanic currents and local conditions and adaptations may play a role in shaping Draconematidae distributions. This highlights the need for further interdisciplinary approaches to fully elucidate the evolutionary and ecological processes governing meiofaunal diversity in deep-sea environments.

## Inactive vent structure remarks and conservation implications

Hydrothermal vents have been the focus of numerous ecological studies since their discovery in 1977. These investigations have significantly enhanced our understanding of the structure and dynamics of benthic communities and the role of environmental conditions at various spatial and temporal scales (*Godet, Zelnio & Van Dover, 2011*). These habitats are known for their unique biogeochemical characteristics, which include commercially valuable mineral resources such as iron, copper, and zinc (*Van Dover, 2019*). However, the prospect of mining these sites poses serious environmental threats, including

permanent alterations in the local topography and removal of habitats (*Boschen et al., 2013*). Furthermore, mining could release toxic metals, disrupt ecological functions, and hinder the recruitment and recovery of sessile invertebrates, particularly in regions where hard substrata are limited (*Van Dover, 2019*).

While the fauna of active sites has been the focus of most vent studies, that of inactive sites is virtually unknown. Few studies report the presence of filter-feeders on the relief created by these mineral-rich mounds (*Boschen et al., 2013*; *Van Dover, 2019*). Moreover, it is suspected that these mineral deposits may host totally different communities than those found at active vents. This is supported by an eDNA study by *Cowart et al. (2020)* on the Lucky Strike vent field, which observed significantly higher diversity (OTUs—Operational Taxonomic Units) in both inactive and peripheral regions compared to active ones, as well as notably distinct communities among the active, inactive, and peripheral areas. While inactive vent systems differ from active ones, both of them face significant threats from deep-sea mining, with potentially severe consequences for biological communities and ecosystem functioning. This challenge is exacerbated by the limited knowledge on these habitats and their associated communities, particularly meiofaunal organisms, which are often overlooked in ecological studies. These knowledge gaps may hinder the development of effective management and conservation strategies (*Menini et al., 2023*).

By documenting the species present at inactive vents, researchers can better assess their ecological roles and connections with neighboring active systems. The discovery of *Dracograllus miguelitus* **sp. nov.** at an inactive vent structure exemplifies the biodiversity hidden in these understudied environments and highlights the urgent need for species-level research. Such findings are crucial for balancing conservation priorities with industrial ambitions, ensuring that management strategies are grounded in a comprehensive understanding of ecosystem dynamics and connectivity.

## CONCLUSIONS

The discovery of a new nematode species not only provides valuable taxonomic and ecological data on a poorly studied genus but also underscores the ecological significance of inactive hydrothermal structures. These habitats increasingly warrant attention in the face of deep-sea mining threats. Future research should aim to further investigate the biodiversity and ecological roles of nematodes and other meiofauna in inactive vent ecosystems, integrating these findings into conservation and management strategies.

While these findings advance our understanding of vent nematode biodiversity, the study is limited to a single structure. Broader exploration across diverse hydrothermal regions and inactive structures and areas is essential to fully understand the genus distribution, biogeography, and ecological roles. Notably, the presence of *D. miguelitus* **sp. nov.** on an inactive structure may result from dispersal events from nearby areas. This highlights specific adaptations to both substratum type and heterogeneity, as well as hydrothermal influences, which require further in-depth study. In conclusion, this study emphasizes the importance of incorporating species-level data into hydrothermal vent research and highlights the urgent need for proactive conservation measures to safeguard

the biodiversity of all types of hydrothermal habitats in the face of increasing anthropogenic pressures.

## Dichotomous key to *Dracograllus* valid species

The dichotomous key was constructed based on previous studies (*Allen & Noffsinger, 1978*; *Decraemer, Gourbault & Backeljau, 1997*; *Min et al., 2016*). The complete list for the description of valid *Dracograllus* species is listed in the reference section.

**1.** Four CATs on rostrum ..................................... ***D. miguelitus* sp. nov**.

   - More than four CATs on rostrum .............................................. 2

**2.** Without sublateral cephalic acanthiform setae on rostrum .......................... 3

   - With one pair of bilateral cephalic acanthiform setae on mid-rostrum
.................................................................. ***D. stekholveni***

**3.** Males with 3–4 preanal corniform setae; 10–11 CATs. Females with 10 CATs: SlATn 18, SvATn 13–16 ...................................................... ***D. cornutus***

   - Males without preanal corniform setae; more CATs. Females with larger numbers of PATs .............................................................. 4

**4.** Males with 7–8 short stiff setae in subventral rows just anterior to SvATl; spicules 39 μm long. Females with 24 SlAT, including 2 tubes posterior to the anus. Both sexes with a swollen pharyngeal region representing 22% of the total body length ... ***D. gerlarchi***

   - Males with 3–4 short stiff setae in subventral rows just anterior to SvATl; spicules 71 μm long. Females with 21 SlAT, including 3 tubes posterior to the anus. Swollen pharyngeal region 13–14% of total length .............................. ***D. falcatus***

**5.** Six CATs on rostrum; males with 5 SlAT, 2–3 SvAT; total length 290 μm; spicules 18 μm ................................................................ ***D. minutus***

   - Eight CATs on the rostrum. Number of PAT higher in males; spicule typically long ........................................................................ 6

**6.** All CATs adjacent to or posterior to the amphideal fovea .......................... 7

   - All CATs anterior to the amphideal fovea .................................. ***D. eira***

**7.** Males with preanal corniform setae; slender, conspicuously long and short SlAT alternating in both sexes ......................................... ***D. trispinosum***

   - Males without preanal corniform setae; SlAT without alternating long and short tubes in both sexes ......................................................... 8

**8.** Several somatic setae in the posterior body region with spiny cuticular insertion and non-annulated tail tip representing 59% of tail length ................... ***D. spinosus***

   - Somatic setae with spiny insertion collar ...................................... 9

**9.** Some somatic setae pedicellate; pedicels 1–8 μm long ........................... 10

   - Somatic setae without pedicels ............................................. 13

**10.** Males with 5–9 SlAT; females with 6–12 SlAT, all anterior to the anus; 9–14 SvAT ....................................................................... 11

    - Males with 12–24 SlAT; females with 15 SlAT (1 posterior to anus) and 16 SvAT.................................................................... ***D. mawsoni***

**11.** Males with 5–7 SlAT; females with 9–13 SvAT.................................... 12

    - Males with 9 SlAT; females with 14 SlAT ................................. ***D. cobbi***

**12.** Males and females with pedicellate setae in ventrosublateral row just anterior to SlAT; spicules 45–53 µm; females with 6–8 SlAT ............................. ***D. demani***

    - Without pedicellate setae in ventrosublateral rows; spicules 36 µm; females with 12 SlAT ........................................................................ ***D. kreisi***

**13.** Annulated body cuticle without ornamentation.................................... 14

    - Annulated body cuticle ornamented with spines, dots, and vacuoles.............. 15

**14.** Amphids long, inverted U-shaped in both sexes; males with 10 SlAT, 11 SvAT; spicules 29 µm; females with 11–13 SlAT and 9–11 SvAT...................... ***D. papuensis***

    - Amphids sexually dimorphic: loop-shaped in males, elongated unispiral in females; males with 7 SlAT, 11 SvAT; spicules 46 µm; females with 8–11 SlAT and same for SvAT ....................................................................... ***D. solidus***

**15.** Body cuticle with vacuolar and granular ornamentation .................***D. wieseri***

    - Body cuticle ornamented with dots and spines.................................... 16

**16.** Body annules ornamented with two rows of dots ................................ 17

    - Spiny ornamentation of the body cuticle ........................................ 18

**17.** Amphids long, oblique loop-shaped in females; tail slender (tail/abd = 5.6)
................................................................... ***D. chitwoodi***

    - Amphids inverted U-shaped in females; tail/abd = 3.9 ................... ***D. timmi***

**18.** Amphids long, inverted U-shaped, as long as the rostrum........................ 19

    - Amphids short and wide, inverted U-shaped ........................ ***D. antillensis***

**19.** Short body (L = 310 µm); faint rostrum ornamentation; body annules with minute spines; spicule 26 µm; males with 6 SvAT ............................... ***D. pusillus***

    - Body > 400 µm; spiny rostrum ornamentation; longer spicules; more than 6 SvAT in males............................................................................. 20

**20.** Long swollen pharyngeal region; amphids inverted U-shaped in males and elongated unispiral in females .................................................***D. laingensis***

    - Shorter, wider swollen pharyngeal region; amphids U-shaped in both sexes; spicules 68 µm .................................................................***D. grootaerti***

**21.** Body annules with dot-like punctations; no anal flap; females with two pairs of paravulvar setae (anterior and posterior to vulva, 5–6 µm) .............. ***D. filipjevi***

    - Anal flap present; females with different number/position of paravulvar setae.... 22

**22.** Large spaces between body annules; females without paravulvar setae ..***D. gilbertae***

    - Body annules closely spaced; females without paravulvar setae.................... 23

**23.** Body annules without ornamentation; paravulvar setae absent ........... ***D. ngakei***

    - Paravulvar setae present; body annules with ornamentation ..................... 24

**24.** Body annules with numerous ridges and spiny protrusions, denser in lateral fields. Males with large loop-shaped amphids with ventral branch longer than dorsal, extending to the first body annule ..................................... ***D. trukensis***

## Abbreviations

The terminology used for the description and measurements was according to *Min et al. (2016)*, *Leduc & Zhao (2016)* and the classical approach by *de Man (1880)*. The abbreviations used are as follows:

| | |
|---|---|
| **L** | body length |
| ***a*** | ratio body length/body maximum width |
| ***b*** | ratio body length/pharynx length |
| ***c*** | ratio body length/tail length |
| ***c'*** | ratio tail length/anal body diameter |
| **C. m.** | constrictor muscles of the vulvar region |
| **abd** | anal body diameter |
| **Amphid from ant** | distance from amphid to anterior end |
| **Amphid. Length** | amphideal length |
| **Amphid. Width** | amphideal width |
| **Amphid./cbd (%)** | amphideal length relative to the corresponding body diameter in % |
| **CATs** | cephalic adhesive tubes |
| **CATl** | length of cephalic adhesive tubes |
| **CATn** | number of cephalic adhesive tubes |
| **Gub. Apoph. Length** | gubernaculum apophysis length |
| **Head diam.** | head diameter |
| **Max. body diam. Mb.** | maximum body diameter at mid-body region |
| **Max. body diam. Phar.** | maximum body diameter at pharynx level |
| **Min. body diam. Mb.** | minimum body diameter at mid-body region |
| **Non. ann. T. length** | non-annulated tail tip length |
| **Non. ann. T. length (%)** | percentage of non-annulated tail tip relative to total tail length |
| **Ov.** | ovaries |
| **PATs** | posterior adhesive tubes |
| **Phar. bulb diam. (ant.)** | anterior pharyngeal bulb diameter |
| **Phar. bulb diam. (post.)** | posterior pharyngeal bulb diameter |
| **Phar. length** | pharynx length |
| **PS** | paravulvar setae |
| **PS length** | paravulvar setae length |
| **SlAT** | sublateral adhesive tubes |
| **SvAT** | subventral adhesive tubes |
| **SlATn** | number of sublateral adhesive tubes |
| **SvATn** | number of subventral adhesive tubes |

| | |
|---|---|
| **1SlAT1** | length of first sublateral adhesion tubes |
| **1SvATl** | length of first subventral adhesion tube |
| **Spic** | spicule |
| **Spic. Length** | spicule length (measured along the median line) |
| **T. length** | tail length |
| **T. length/abd** | ratio of tail length to anal body diameter |
| **V. (%)** | vulva position as percentage of total body length |
| **V. b. diam.** | vulvar body diameter |
| **V. to ant.** | distance from vulva to anterior end |

## ACKNOWLEDGEMENTS

We thank the crew of the R/V L'Atalante and the pilots of the ROV Victor6000, as well as the chief scientist of the MoMARSAT 2018 cruise (DOI: 10.17600/18000514). We also express our deep gratitude to Daniel Leduc (National Institute of Water and Atmospheric Research—NIWA, New Zealand) for his valuable comments on Draconematidae identification.

### Funding

This study was conducted within the context of the European Deep-Rest project (Conservation & Restoration of Deep-Sea Ecosystems in the Context of Deep-Sea Mining, https://deep-rest.ifremer.fr/), which aims to enhance our capacity for science-based spatial planning and ecosystem management in areas threatened by deep-sea mining, such as hydrothermal vents and polymetallic nodules. Deep-Rest project was funded through the 2020–2021 Biodiversa and Water JPI joint call for research projects, under the BiodivRestore ERA-NET Cofund (GA N°101003777), with the EU and the following funding organisations: Agence Nationale de la Recherche (ANR-21-BIRE-0003), France, Ministry of Agriculture, Nature and Food Quality (LNV), Netherlands, Research Foundation—Flanders (FWO), Belgium, German Federal Ministry of Research (BMBF) through VDI/VDE-IT, Germany, Environmental Protection Agency (EPA), Ireland, Fundação para a Ciência e a Tecnologia (FCT), Portugal, Fundo Regional para a Ciência e Tecnologia (FRCT), Portugal-Azores and State Research Agency (AEI), Spain. William Johnson da Silva's PhD research was entirely funded by DEEP REST. We also received support from the Meiodyssea (Massive mEIOfauna DiscoverY of new Species of our oceans and SEAs) funded by the Ocean Shot Research Grant Program of the Sasakawa Peace Foundation supported by the Nippon Foundation and the Blue Revolution (Biodiversity underestimation in our bLUe planEt: artificial intelligence REVOLUTION in benthic taxonomy (https://bluerevolution.ifremer.fr/fr)) projects, which contributed to image processing and species identification. The funders had no role in study design, data collection and analysis, decision to publish, or preparation of the manuscript.

## Grant Disclosures

The following grant information was disclosed by the authors:

European Deep-Rest Project.

2020–2021 Biodiversa and Water JPI Joint Call for Research Projects.

BiodivRestore ERA-NET Cofund: GA N°101003777.

Agence Nationale de la Recherche, France: ANR-21-BIRE-0003.

Ministry of Agriculture, Nature and Food Quality (LNV), Netherlands.

Research Foundation—Flanders (FWO), Belgium.

German Federal Ministry of Research (BMBF) through VDI/VDE-IT, Germany.

Environmental Protection Agency (EPA), Ireland.

Fundação para a Ciência e a Tecnologia (FCT), Portugal.

Fundo Regional para a Ciência e Tecnologia (FRCT), Portugal-Azores.

State Research Agency (AEI), Spain.

DEEP REST.

Meiodyssea (Massive mEIOfauna DiscoverY of new Species of our oceans and SEAs) funded by the Ocean Shot Research Grant Program.

Sasakawa Peace Foundation.

Nippon Foundation.

Blue Revolution.

## Competing Interests

The authors declare that they have no competing interests.

## Author Contributions

- William Johnson da Silva performed the experiments, analyzed the data, prepared figures and/or tables, authored or reviewed drafts of the article, and approved the final draft.
- Daniela Zeppilli analyzed the data, authored or reviewed drafts of the article, and approved the final draft.
- Valentin Foulon performed the experiments, analyzed the data, prepared figures and/or tables, authored or reviewed drafts of the article, and approved the final draft.
- Pierre-Antoine Dessandier analyzed the data, authored or reviewed drafts of the article, and approved the final draft.
- Marjolaine Matabos conceived and designed the experiments, authored or reviewed drafts of the article, and approved the final draft.
- Jozee Sarrazin conceived and designed the experiments, authored or reviewed drafts of the article, and approved the final draft.

## Data Availability

All specimens are registered under MNHN-BN511 at the Muséum National d'Histoire Naturelle (MNHN).

## New Species Registration

The following information was supplied regarding the registration of a newly described species:

Publication LSID: urn:lsid:zoobank.org:pub:AA6564D7-6BA7-405E-94D3-B659E62B8BDB

Dracograllus miguelitus sp. nov. LSID: urn:lsid:zoobank.org:act:66AF55AB-13DC-469A-A212-BDA20B4699BB

## Supplemental Information

Supplemental information for this article can be found online at http://dx.doi.org/10.7717/peerj.19585#supplemental-information.

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
