# Peer review of "On *Dracograllus miguelitus* sp. nov. (Nematoda: Draconematidae) from an inactive structure: insights into its taxonomy, biodiversity and ecology at hydrothermal vents"

_PeerJ, doi:10.7717/peerj.19585_

## Round 0.1 · original submission · Minor Revisions

Dear Mr. Johnson da Silva,

Your manuscript has now been evaluated. There is still a request for minor changes, before I can accept the manuscript. Please consider the comments carefully and submit the final version as soon as possible.

Reviewer 1 ·

Basic reporting

The manuscript is devoted to the description of the new species of draconeamtidae found in the area of hydrotarmal vents. The nmanuscipt it written in the perfect English. Literature is cited in the almost full volume. Figures are of a very good quality.

Experimental design

The research is done in the frame of journal aims and scopes. The modern methods are used.

Validity of the findings

All findings are new to science and the discussion part covers actual topics in geography of free-living marine nematodes.

Additional comments

It was my pleasure to read such a beautiful piece of study. The manuscript is of very good quality and would be intresting to wide range of scientist - from taxonomists to chorologists and ecologists.
It is very pity that there was not possibility to make genetic analisys of described species.
Although, the scaning electron studies of neamtode morphology could provide some additional information on species morphology.
The main question conserning the Comments on the imaging approach part. I belive that the pioneer of confocal microscopy among nematologists was Aldo Zullini who proposed this method in 2006 (ZULLINI, A. & VILLA, A. (2006). Redescription of three tobrilids (Nematoda) from Altherr’s collection using confocal
microscopy. Journal of Nematode Morphology and Systematics 8, 121-132) and deserves to be cited in this manuscript.

·

Basic reporting

The author describes Dracograllus miguelitus sp. nov. from hydrothermal vents of the Mid-Atlantic Ridge as a new species to science. This species is the first within the genus Dracograllus from hydrothermal vent areas and the author discusses the adaptations of this species to the harsh environment of vent fields. This study is important because of the lack of meiofaunal studies in the deep sea. Overall, this paper needs only little improvement. These improvements mainly include the standardization of figures and tables.

Experimental design

I think the experimental design of the study is perfect. The only addition I would recommend is the barcoding of the mitochondrial marker gene COI. Reference libraries are crucial in environmental studies when taxonomic experts are lacking, especially for such small creatures.

Validity of the findings

I honestly cannot comment on the validity of the species description, as I am not at all into nematode taxonomy. But to me the description looks well thought out and tidy. The rules of nomenclature are correctly followed and the key to the genus Dracograllus at the end looks professional.
The findings on hydrothermal vent ecology and adaptations are well interpreted and sound convincing to me.

Additional comments

I have a few comments on the figures, where I would suggest more consistent labelling. Also, some are either too low resolution (Figure 1 & 7) or you can't read the writing in the illustrations (Figure 1 & 7). The figures are the biggest point of criticism in my review - but this should be very easy to fix.
You can find my recommended changes in the PDF file.
Good luck with your PhD!

---

## Round 0.2 · accepted · Accept

In the revised version the authors took into consideration all comments and remarks. I recommend to accept the manuscript for publication in PeerJ.